# Ensemble dynamics and information flow deduction from whole-brain imaging data

**Yu Toyoshima**[1]*, **Hirofumi Sato**[1], **Daiki Nagata**[1], **Manami Kanamori**[1], **Moon Sun Jang**[1¤], **Koyo Kuze**[1], **Suzu Oe**[2], **Takayuki Teramoto**[2], **Yuishi Iwasaki**[3], **Ryo Yoshida**[4], **Takeshi Ishihara**[2], **Yuichi Iino**[1]*

**1** Department of Biological Sciences, Graduate School of Science, The University of Tokyo, Bunkyo-ku, Tokyo, Japan, **2** Department of Biology, Faculty of Sciences, Kyushu University, Nishi-ku, Fukuoka, Japan, **3** Department of Mechanical Systems Engineering, Graduate School of Science and Engineering, Ibaraki University, Hitachi, Ibaraki, Japan, **4** The Institute of Statistical Mathematics, Research Organization of Information and Systems, Tachikawa, Tokyo, Japan

¤ Current address: Neuroscience Institute, Graduate School of Science, Nagoya University, Aichi, Japan
* ytoyo@bs.s.u-tokyo.ac.jp (YT); iino@bs.s.u-tokyo.ac.jp (YI)

**Data Availability Statement:** All supplementary figures and notes are available from DOI (10.6084/m9.figshare.24623805). The datasets of raw time series of neural activity are available from DOI (10.

## Abstract

The recent advancements in large-scale activity imaging of neuronal ensembles offer valuable opportunities to comprehend the process involved in generating brain activity patterns and understanding how information is transmitted between neurons or neuronal ensembles. However, existing methodologies for extracting the underlying properties that generate overall dynamics are still limited. In this study, we applied previously unexplored methodologies to analyze time-lapse 3D imaging (4D imaging) data of head neurons of the nematode *Caenorhabditis elegans*. By combining time-delay embedding with the independent component analysis, we successfully decomposed whole-brain activities into a small number of component dynamics. Through the integration of results from multiple samples, we extracted common dynamics from neuronal activities that exhibit apparent divergence across different animals. Notably, while several components show common cooperativity across samples, some component pairs exhibited distinct relationships between individual samples. We further developed time series prediction models of synaptic communications. By combining dimension reduction using the general framework, gradient kernel dimension reduction, and probabilistic modeling, the overall relationships of neural activities were incorporated. By this approach, the stochastic but coordinated dynamics were reproduced in the simulated whole-brain neural network. We found that noise in the nervous system is crucial for generating realistic whole-brain dynamics. Furthermore, by evaluating synaptic interaction properties in the models, strong interactions within the core neural circuit, variable sensory transmission and importance of gap junctions were inferred. Virtual optogenetics can be also performed using the model. These analyses provide a solid foundation for understanding information flow in real neural networks.

6084/m9.figshare.21968078). All codes of TDE-RICA are available from a GitHub repository at https://github.com/YuToyoshima/TDE-RICA. All codes of gKDR-GMM used for model fitting and plotting are available from a GitHub repository at https://github.com/yuichiiino1/gKDR-GMM.

**Funding:** This work was supported by the CREST programs "Creation of Fundamental Technologies for Understanding and Control of Biosystem Dynamics" (JPMJCR12W1) to and "Establishment of high-speed, high-dimensional closed-loop optical measurement technology and its applications to neuroscience" (JPMJCR22N4) of the Japan Science and Technology Agency (JST) to YI. YI was supported by Grant-in-Aid for Scientific Research, Japan Society for the Promotion of Science (17H06113, 22H00416, 20K21805, 25115009 and 19H04980). YT was supported by JST PRESTO (JPMJPR1947) and Grants-in-Aid for Scientific Research, Japan Society for the Promotion of Science (26830006, 18K14848, 16H01418, 18H04728, 17H05970 and 19H04928). TI was supported by Grant-in-Aid for Scientific Research, Japan Society for the Promotion of Science (20115003, 25115009, 18H05135, 24650167, 19H03326, 17H06113 and 16H0654), JST PRESTO (7700000461), and NTT-Kyushu University Collaborative Research. HS was supported by Grant-in-Aid for Scientific Research, Japan Society for the Promotion of Science (21K15182). The funders had no role in study design, data collection and analysis, decision to publish, or preparation of the manuscript.

**Competing interests:** The authors have declared that no competing interests exist.

## Author summary

Brain is a complex network of interconnected neurons that process sensory and other information through synaptic connections. In this study we measured the activity of all neurons in the head of a nematode worm, *C. elegans*, by using a high-speed fluorescent microscope. Through the application of cutting-edge mathematical methods to the acquired data, we successfully extracted recurring dynamics of subsets of neurons and predicted activities of each neuron. Remarkably, the model autonomously generated network dynamics of the whole brain. By carefully analyzing these reconstructions, neuronal interactions and information flow in the brain could be deduced. Our results present a methodology for understanding the basic construction of brain dynamics through observation of brain activity, which are likely applicable to the brain of other animals.

## Introduction

Understanding how the brain performs various integrative and instructive functions is a fundamental question in neurobiology. The brain's capacity to execute diverse functions depends on the transmission of activity across interconnected networks of neurons. This transmission takes place through chemical and electrical synapses, where signs, efficiency, and dynamic properties of these synapses determine the activity of recipient neurons. Additionally, the interactions between multiple presynaptic inputs received by each neuron play a crucial role in shaping the response of postsynaptic neurons. Therefore, in addition to the intrinsic properties of constituting neurons, the network's shape and synaptic properties are the primary determinants of information flow.

While the mammalian cerebral cortex, composed of tens of billions of neurons, exhibits high complexity, recent technological advancements have begun to reveal connectomes at various levels, ranging from micro-connectome at synaptic resolution to a macroscopic connectome between different brain regions [1–3]. Concurrently, various techniques such as multi-unit electrical recording, electroencephalogram (EEG), magnetoencephalography (MEG), functional magnetic resonance imaging (fMRI) and calcium imaging are actively employed to monitor neuronal activities [4]. Nevertheless, understanding brain dynamics across structural layers, from single neurons to functional brain regions and the whole brain, remains challenging due to the vast number of neurons and the structural complexity of the brain.

In smaller animals, quasi-whole-brain activity measurements at single-cell resolution have shown promise. One of the most successful subjects are zebrafish larvae; for instance, the brain of agar-embedded larval animals has been observed during fictive swimming by light-sheet microscopy, revealing various activity groups and their relationships with behavior [5,6]. Another model animal well-suited for such analyses is the nematode *C. elegans*. With a small body length of about 1 mm in young adults, *C. elegans* allowed for the reconstruction of the entire nervous system and acquisition of the full connectome data through electron microscopy, which includes exactly 302 neurons in adult hermaphrodite [7–9]. "Whole-brain imaging" in *C. elegans* has shown that global neuronal activities form a manifold in the state space, where typical behaviors such as forward, backward and turn are represented [10–13]. Furthermore, whole-brain imaging in freely moving animals has identified activities associated with specific behaviors [14–16], and behaviors were successfully predicted based on whole brain activities [17,18].

Although these attempts have described the whole-brain states and state transitions, and found ensemble activities related to sensory input or behavior, the question of how these

activity patterns are generated has not been directly addressed. This gap in understanding stems from a lack of methodologies for understanding how network dynamics are assembled and how network structure and synaptic connections contribute to the observed dynamics.

In this study, we present new approaches for decomposing and reconstructing the dynamics of the whole nervous system based on *C. elegans* 4D imaging results. These approaches successfully extracted common dynamics across animals such as forward-backward core network, as well as individual differences, especially in the transmission pathways of sensory information. Moreover, the roles of individual neurons and chemical and electrical synapses in the information flow could be estimated by synapse-based models. These findings contribute to deeper understanding of how neural networks generate observed dynamics in the *C. elegans* nervous system.

## Results

### Correlation among neuronal activities

In this study, we recorded neural activities from *C. elegans* adult hermaphrodites expressing the calcium reporter Yellow Cameleon (nuclear localized YC2.60) in all neurons. Each worm was placed in a narrow channel within a microfluidic device known as an olfactory chip [19] and stimulated with a periodic switch between two different NaCl concentrations while under a confocal fluorescence microscope (Fig 1A). The focal plane was scanned up and down to obtain time-lapse 3D fluorescence images, which is called 4D imaging [20–23]. Overall, we obtained a total of 24 whole-brain activity movies at a rate of about 4 volumes per second and corresponding annotation movies (Fig 1B and Materials and Methods).

Fig 2A and S1 Text—File 1 illustrate some noteworthy characteristics in the neuronal activities obtained by our 4D imaging. First, as the animals received sensory stimuli (changes in NaCl concentration applied to the nose tip), small groups of neurons displayed responses to the sensory stimuli. Other groups of neurons exhibited synchronized activation and inactivation, but their activity patterns did not show clear synchrony with the sensory stimuli and were considered spontaneous activities. These observations are in line with previous findings [10].

To further clarify these relationships, we calculated cross-correlations in neuronal activities for all pairwise neuron combinations (Fig 2B). The size of correlated groups varied considerably between individual samples. However, in many cases, there was at least one major group of neurons with correlated activity and another correlated group negatively correlated with the first group. Upon examining the member neurons within these groups, we confirmed that they correspond to well-known groups of neurons related to reversal and forward movements, namely, AVA, AVE, and RIM neurons in the first group (called group A here), and RIB, RID, and RME neurons in the second group (called group B). Additionally, we identified other groups that showed correlations across samples, including (OLLL, OLLR, OLQDL, OLQDR) and (I2L, I2R, MCL), as well as the left and right members of BAG, RIA, RMDV and SMDD classes (Fig 2C).

### Common and individual dynamics revealed by time-delay embedding and independent component analysis

Next, we focused on the overall dynamics of the nervous system as captured by the whole-brain activity data. In previous studies, principal component analysis (PCA) was utilized for analyzing the whole-brain activity data of individual samples [10,18]. However, this approach has some drawbacks.

## A

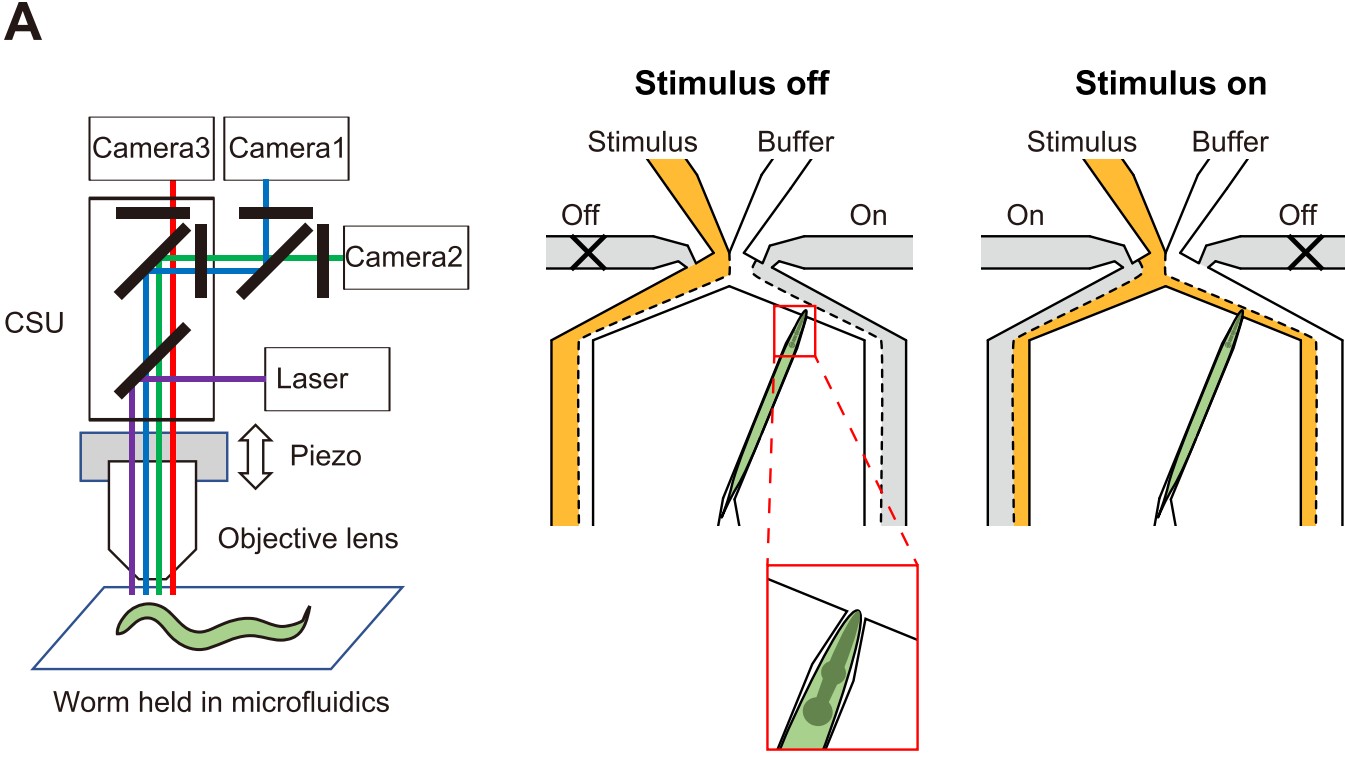

## B

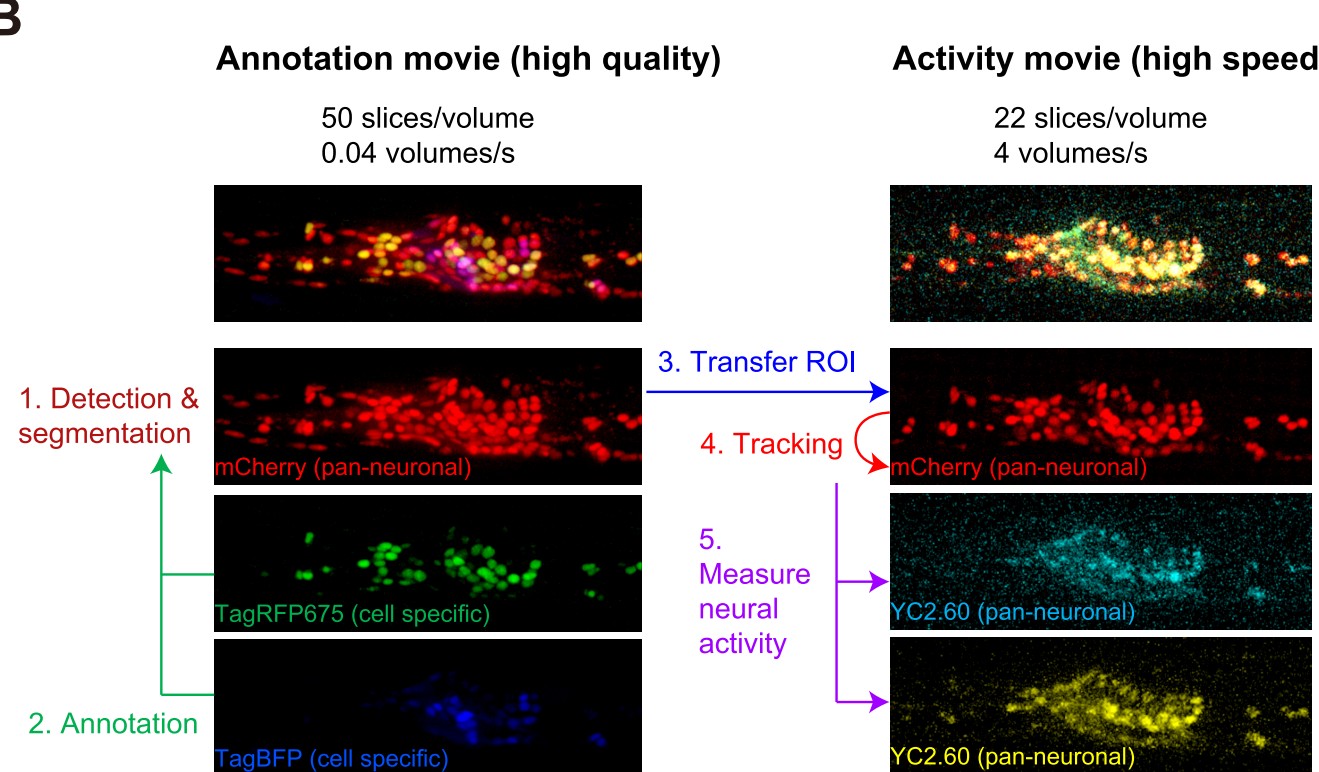

**Fig 1. Experimental setup for 4D imaging.** (A) Left panel shows an overview of the microscope for 4D imaging. The head of the worm was observed by the spinning-disk confocal microscope with 3 cameras for simultaneous multi-color imaging and with the piezo-positioner for z-scanning. Right panel shows that the worm was held in the microfluidic olfactory chip and was stimulated periodically. The inset shows the slight movement of the worm in the chip. CSU:

confocal scanning unit. (B) An overview of the image analysis pipeline. The worm expressed 5 nuclear-localized fluorescent proteins and 3 colors were recorded for the annotation movie, and after that 3 colors were recorded for the activity movie. The neuronal nuclei were detected in the high-quality annotation movie. The nuclei were then annotated by labeling them with the names of their respective neurons. Next, the ROI information of the nuclei in the annotation movie was transferred to the activity movie. The nuclei were tracked in the activity movie and neural activity was calculated. ROI: region of interest.

The first drawback is that the principal component (PC) axes obtained from one sample cannot be directly matched to those of other samples, hindering meaningful cross-sample comparison of the dynamics in the PCA space. Secondly, PCA requires orthogonality between PC axes, which is a mathematical constraint that may lack biological validity and distort the obtained components. This requirement can be relaxed by using independent component analysis (ICA) instead of PCA [24]. Lastly, while PCA is useful for decomposing the whole-

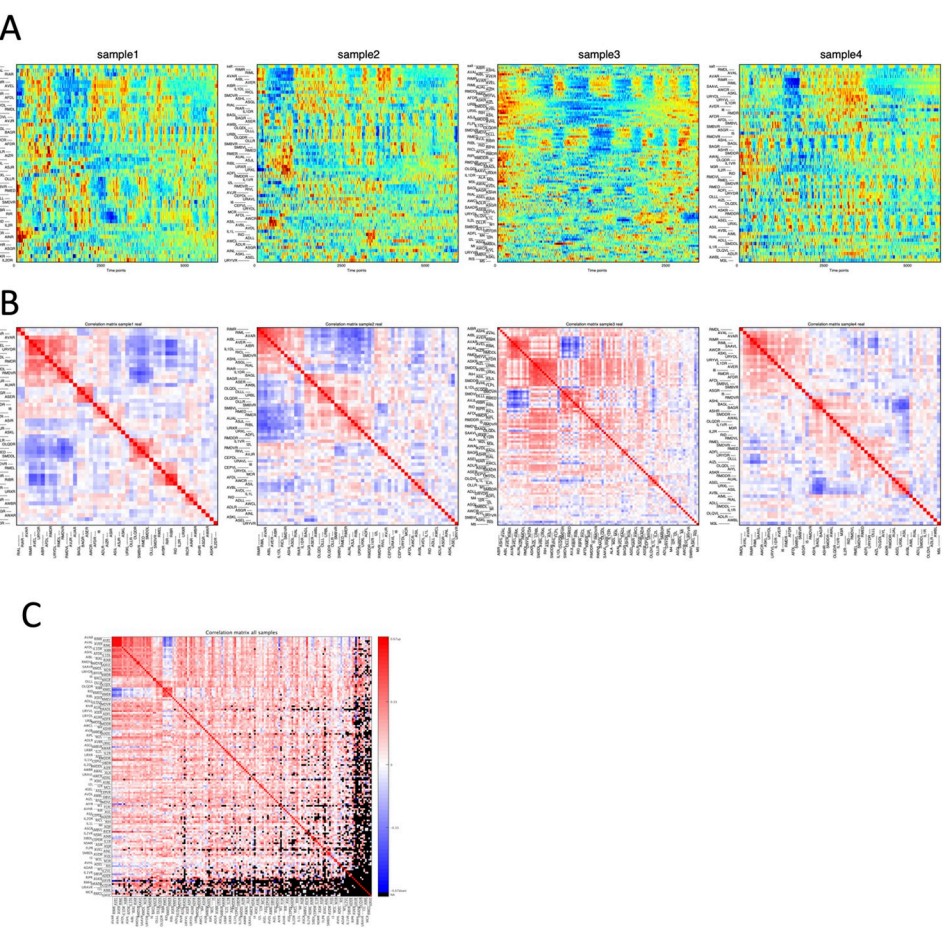

**Fig 2. Data obtained by 4D imaging.** (A) Activity time series of head neurons obtained by 4D imaging. The activity of each neuron in the scaled fluorescence ratio of YFP over CFP is shown in pseudocolor. The top row shows the salt concentration applied to the animal's nose, and remaining rows indicate the neuronal activity profiles. Each row represents one neuron, whose order was determined by hierarchical clustering based on activity cross-correlations. Note that only a subset of head neurons, which differ between samples, are shown in each panel, because some neurons were unobserved (*e.g.*, too dim) or unannotated. (B) Pairwise cross-correlation of head neuron activities obtained by 4D imaging. Red and blue color show positive and negative correlation, respectively. In (A) and (B), examples of four samples are shown. For all samples, see S1 Text—File 1. (C) Pairwise cross-correlation averaged across samples. Hierarchical clustering based on p-values was performed to arrange the neurons. Red and blue color show positive and negative correlations, respectively. Several large and small correlated groups are observed; two showing prominent negative correlation with each other. Pairs of neurons never co-observed in any sample are filled in black.

brain dynamics into instantaneous components, it is not suitable for extracting the "temporal motifs" of neural activities from the time series data. A temporal motif is a latent temporal pattern that appears repeatedly in neural activities.

To address these limitations, time-delay embedding (TDE) methods have been applied to model complex ecological systems [25] and neural dynamics [26]. Particularly, a method combining TDE with ICA was applied to extract behavioral motifs from a time series of worms' postures [27]. A similar approach should be useful for extracting neural motifs from whole-brain activity data.

Here we developed a method called Reconstruction ICA with time-delay embedding (TDE-RICA) that is applicable to the whole-brain activity data from multiple samples (Fig 3). RICA, a variant of ICA, introduces penalties for reconstruction costs, ensuring that the captured components and weights effectively reproduce the original data [28]. S1 Text—File 2 provides an in-depth illustration of how TDE-RICA extracts typical temporal dynamics (motifs) from time series data.

The presence of unidentified neurons in each sample (as a consequence of the challenges in perfect neuron identity annotation; [23]) led to the occurrence of missing values in our dataset. Because RICA by itself cannot handle missing values, we selected 94 neurons of 10 samples, without any missing values among them, from the entire dataset of 177 neurons from 24 samples (which include missing values).

To capture the long-time-scale dynamics such as switching of forward and backward command neurons [10], we set the time-delay for embedding as 300 time steps (approximately 60 s). This corresponds to the duration of the NaCl stimulation period used in the experiment (approximately 60 s). The whole-brain activity was measured during 6000 time steps (approximately 1200 s), and the embedded data is represented by a matrix $X$ containing (94 [neurons] $\times$ 300 [delay time steps]) $\times$ (5701 [time steps] $\times$ 10 [samples]) elements (see Methods and Fig 3). We set the number of components as 14, which was the minimum number required to capture the neural response to the NaCl stimulation (see Methods). Applying TDE-RICA to the selected dataset returns the independent components $M$ ($M = W^T X^T$, matrix $M$ contains 14 [components] $\times$ (94 [neurons] $\times$ 300 [delay time steps]) elements) with the corresponding weight matrix $W$ (containing (5701 [time steps] $\times$ 10 [samples]) $\times$ 14 [components] elements). These components were estimated through numerical optimization to effectively represent the original data (Fig 4A, for all 10 selected samples, see S1 Text—File 3). The estimated independent components ($M$) can be considered motifs of neural activities shared across samples, while estimated weights ($W$) represent sample-specific motif occurrences.

To obtain the activity motifs for all neurons, including those with missing values, we extended the results obtained using TDE-RICA as above with a matrix factorization. This technique efficiently completed missing values. By using the matrices of motifs and their occurrences obtained from the partial dataset without missing values, we expanded the matrix of motifs to all 177 neurons and the matrix of motif occurrences to all 24 samples (see Fig 3). This allowed the product of motifs and their occurrences to accurately reproduce the observed neural activities. Consequently, we obtained the full set of motifs ($M^{all}$ containing 14 [components] $\times$ (177 [neurons] $\times$ 300 [delay time steps]) elements) and their occurrences ($W^{all}$, containing (5701 [time steps] $\times$ 24 [samples]) $\times$ 14 [components] elements) (S1 Text—File 4).

The motifs and their occurrences obtained by TDE-RICA from the partial dataset were well preserved and extended through matrix factorization. The analyzed results of the full set of motifs and occurrences (see S1 Text—File 5) were consistent with those of the motifs and the occurrences obtained by TDE-RICA from the partial dataset (see S1 Text—File 3). Therefore, we only describe here the results from the full set of motifs and occurrences.

**A  Time-Delay Embedding (TDE)**

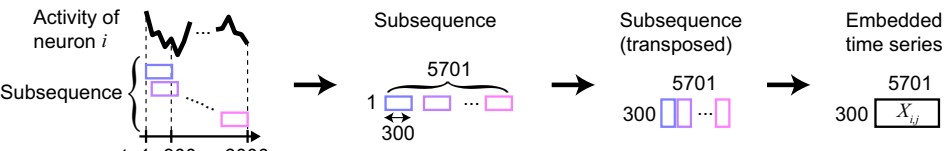

**B  Assembling data**

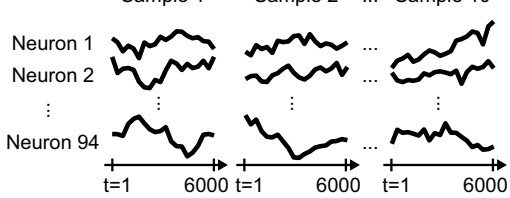

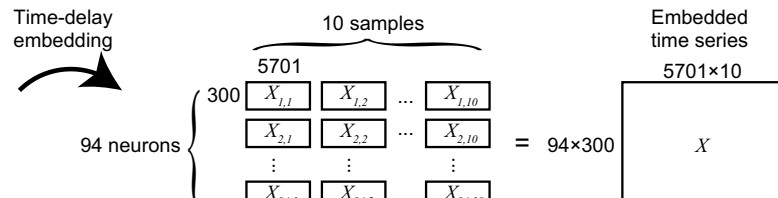

**C  Reconstruction Independent Component Analysis (RICA)**

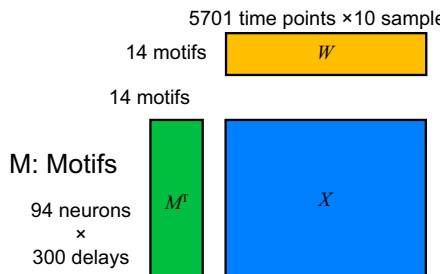

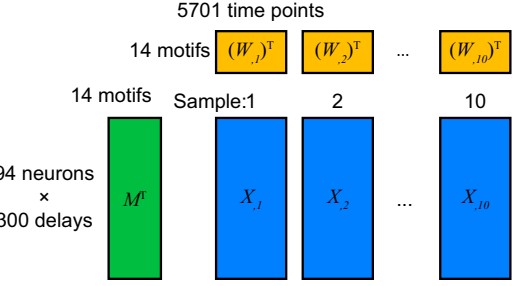

**D  Reconstruction Independent Component Analysis with Matrix Factorization (RICA-MF)**

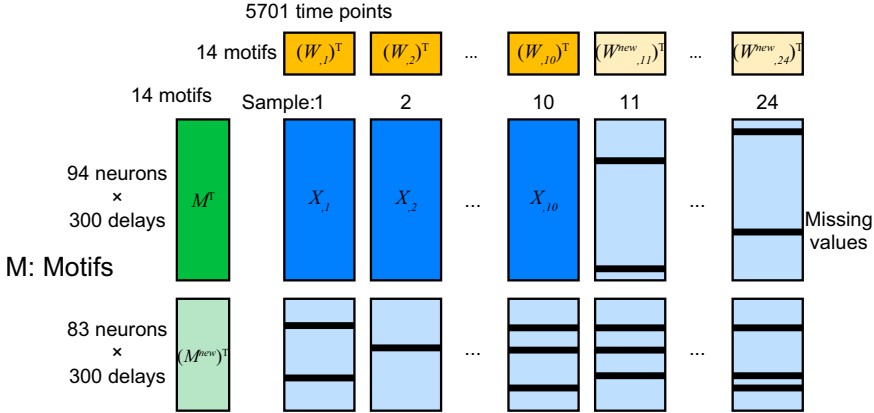

**Fig 3. Graphical representation of TDE-RICA and matrix factorization procedure.** (A and B) Principles of time-delay embedding (TDE) of time-series data are graphically depicted. (C) Principles of reconstruction independent component analysis (RICA), used for a subset of data with no missing values, are graphically depicted. (D) Principles of matrix factorization (MF) used for all neurons in all samples, with missing data, are graphically depicted.

The number of time-delay steps for TDE-RICA affect the analyzed results including motifs and occurrences. We tested the effect of time-delay steps by applying TDE-RICA at different time-delay steps to simple time series data (S1 Text—File 6). The results show that the number of time delay steps appropriate for robust results ranges from half to the same of the typical frequency of the original time series. However, these findings are based on simple time series and do not always hold true for more complex data including the whole-brain neural activity, which contains multiple typical frequencies. We therefore applied TDE-RICA to the whole-brain neural activity dataset with different time-delay steps (see S1 Text—File 7). We found that the obtained motifs and occurrences are somewhat consistent among several different numbers of time-delay steps. Therefore, we only describe here the results when the number of time-delay steps is set to 300.

If the sensory neurons directly control the downstream neurons in response to NaCl stimulation, the motifs corresponding to the sensory neurons are expected to include the activity of downstream neurons as well. We found that the last two out of the 14 motifs (13th and 14th) had large weights on neurons responding to the sensory stimulation, including ASE and BAG sensory neurons (Figs 4B and 5A). Other neurons, including downstream interneurons, had lower weights, suggesting that activities of sensory neurons might only weakly influence the downstream neurons.

To investigate the effects of sensory stimulation on neural activity, we visualized the changes in neural activity induced by NaCl stimulation (S1 Text—File 8). Sensory neurons including ASEL/R and BAGL/R showed significant response to the stimuli on average. Interneurons and motor neurons also responded to the stimuli in some cases, but in others they did not respond or responded in the opposite direction, showing large individual differences. Such variability diminishes the average response of the neurons. Even in neurons that responded on average, the magnitude of the response was small compared to the overall variance. These trends on average are consistent with the results of TDE-RICA, suggesting that this analysis successfully extracted common behaviors among individuals as motifs. In addition, considering that several neurons involved in forward and backward movement showed clear spontaneous and synchronous activity, it would be difficult to predict the activity of these neurons based on sensory input alone.

We also found that the 1st and 2nd motifs had large weights on neurons governing worms' forward and backward movements, such as interneurons AVA, AIB, RIM, AVB, and RME motor neurons (Fig 5B). The activities of neurons governing forward (AVB and RME, group B) and backward movements (AVA, AIB, RIM, group A) were inversely correlated, suggesting the presence of mutual inhibition mechanisms [29]. Interestingly, the activity of mechanosensory neurons including OLQ and OLL was positively correlated to that of the neurons governing forward movements in these motifs. This might suggest that when worms move forward in the microfluidic device, their heads inevitably experience mechanical stimulation from the device's wall.

In the 8th motif, the neurons involved in mechanosensation and backward movements displayed large negative weights, and those for forward movement had smaller weights (Fig 5C). This indicates variable and context-dependent relationships between these three groups of neurons. Notably, in this motif, the thermosensory neurons AFD and their downstream interneurons RIA and RMD exhibited positive correlation with the neurons for backward

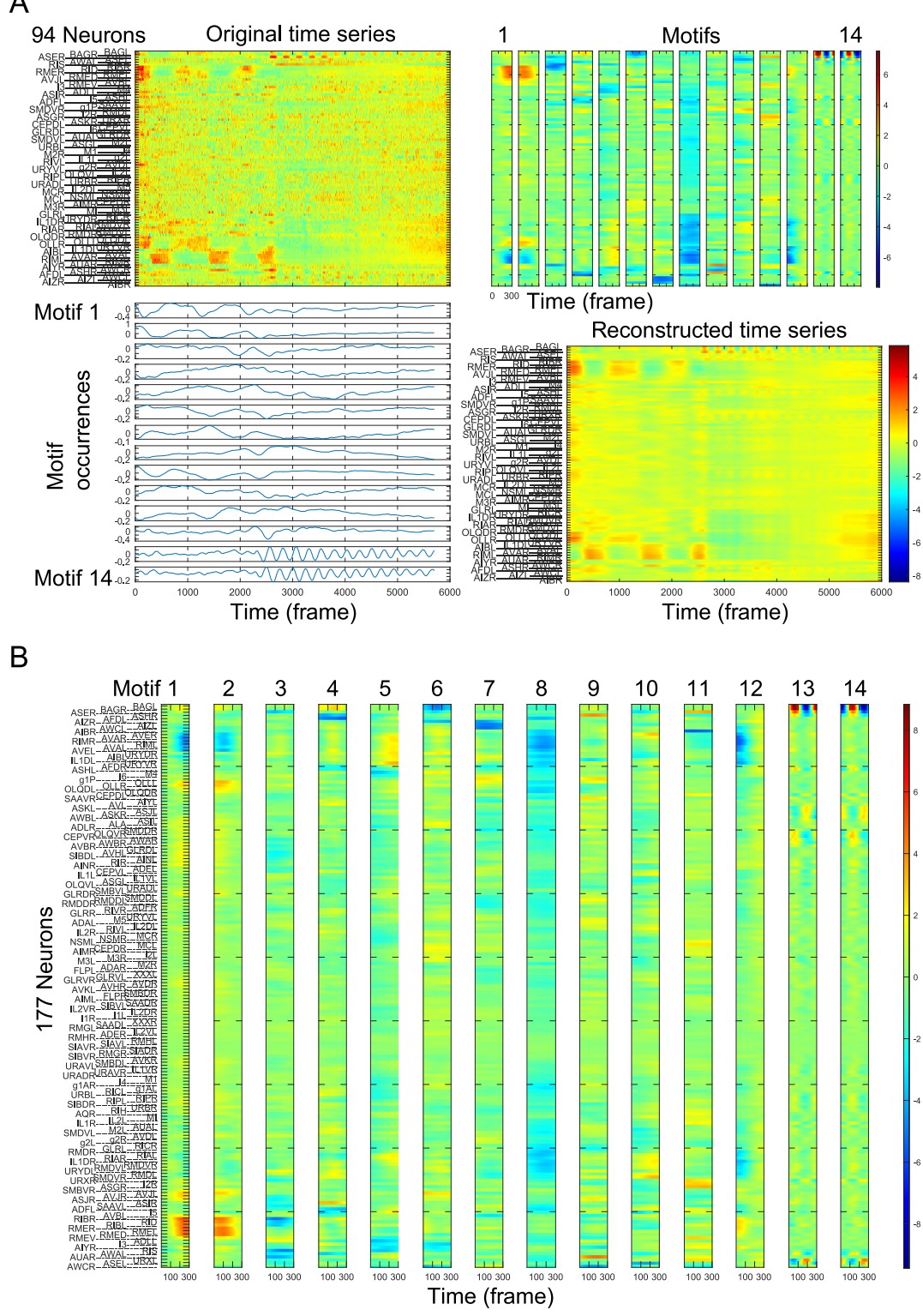

**Fig 4. TDE-RICA captures motifs of neural activity.** (A) TDE-RICA for neurons commonly observed across samples. (A: Top left) Original time series data shown as a heat map as Fig 2A. (A: Top right) Motifs of neural activities obtained by TDE-RICA. Each motif consists of the activity of 94 neurons over 300 time points. The color indicates the relative intensity of each individual neural activity in each motif. Motifs are common between samples. (A: Bottom left) Motif occurrences obtained by TDE-RICA. The occurrences differ between samples. (A: Bottom right) Reconstructed time series. The color

range is the same as that of the upper left panel. (B) Fourteen motifs are shown, each consisting of the activity of 177 neurons over 300 time points. The color indicates the relative intensity of each individual neural activity per motif.

movements [30,31]. In addition, chemo/thermosensory neurons AWC and downstream interneurons AIZ were positively correlated in the 7th motif. These results suggest that these motifs may have captured the unintended activities of the thermotaxis circuit.

To further explore the relationships between motifs, we analyzed the occurrences of the motifs by calculating the cross-correlations of the occurrences in each sample and averaging them across samples (S1 Text—File 9).

We found a prominent cross-correlation between the 13th and 14th motifs that represent the sensory responses. To compare the neural dynamics between different samples in the latent space of the motifs, we plotted the occurrences of the 13th motif along with that of the 14th motif (Fig 5D). The 14th motif always preceded the 13th motif. In the phase diagram, their trajectories formed a circle during the period with sensory stimulation, which degenerated to the origin during the period without stimulation. These results suggest that the sensory motifs successfully represent the presence or absence of stimuli and capture the response to the periodic stimulus. These trajectories were common between samples, indicating that these sensory dynamics are common across animals.

Furthermore, we found another prominent cross-correlation between the 1st and the 2nd motifs that represent the forward and backward movements (Fig 5E). The first motif always preceded the second motif. In the phase diagram, the trajectories formed distorted circles. Although these features were common between samples, the shapes of trajectories were different. This indicates individual differences in the quantitative dynamics of command interneurons and motor neurons.

Additionally, we identified an interesting relationship between the 8th and 9th motifs. These motifs were positively correlated in one sample but negatively correlated in another (Fig 5F). These relationships remained consistent throughout the recording of each sample. This result suggests individual differences in the qualitative relationships of neurons between animals, despite the general assumption of little individual differences in neural activity in *C. elegans* due to their uniform genetic background and stereotypic developmental process [9].

## The synapse-based model reproduces the overall brain network dynamics

The TDE-RICA successfully decomposed the overall dynamics of the *C. elegans* nervous system into the separate dynamics of neuron subsets, represented by motifs. Our next objective was to understand the synaptic basis of the dynamics. To achieve this, we utilized the connectome information in the analysis of the observed neuronal activities. As described in the Introduction, the synaptic connections between neuron pairs throughout the nervous system (connectome) have been fully described. Based on this information, we constructed models in a bottom-up manner.

In these models, we only considered annotated neurons with non-random time series activity (time series with considerable autocorrelation). Unlike TDE-RICA, each sample was treated separately but not combined (see Materials and Methods). First, in a selected sample (sample *j*), we selected one neuron as the "target" neuron (neuron *i*). We then identified neurons that send synaptic inputs to the target neuron via chemical or electrical synaptic connections (presynaptic neurons or "explanatory neurons" for target *i*). Our assumption is that the activity of the target neuron is determined by inputs from presynaptic neurons, and thus, we treated chemical synapses as directed and electrical synapses as undirected. Time-delay embedding was applied to the explanatory neurons to predict the target neuron's activity

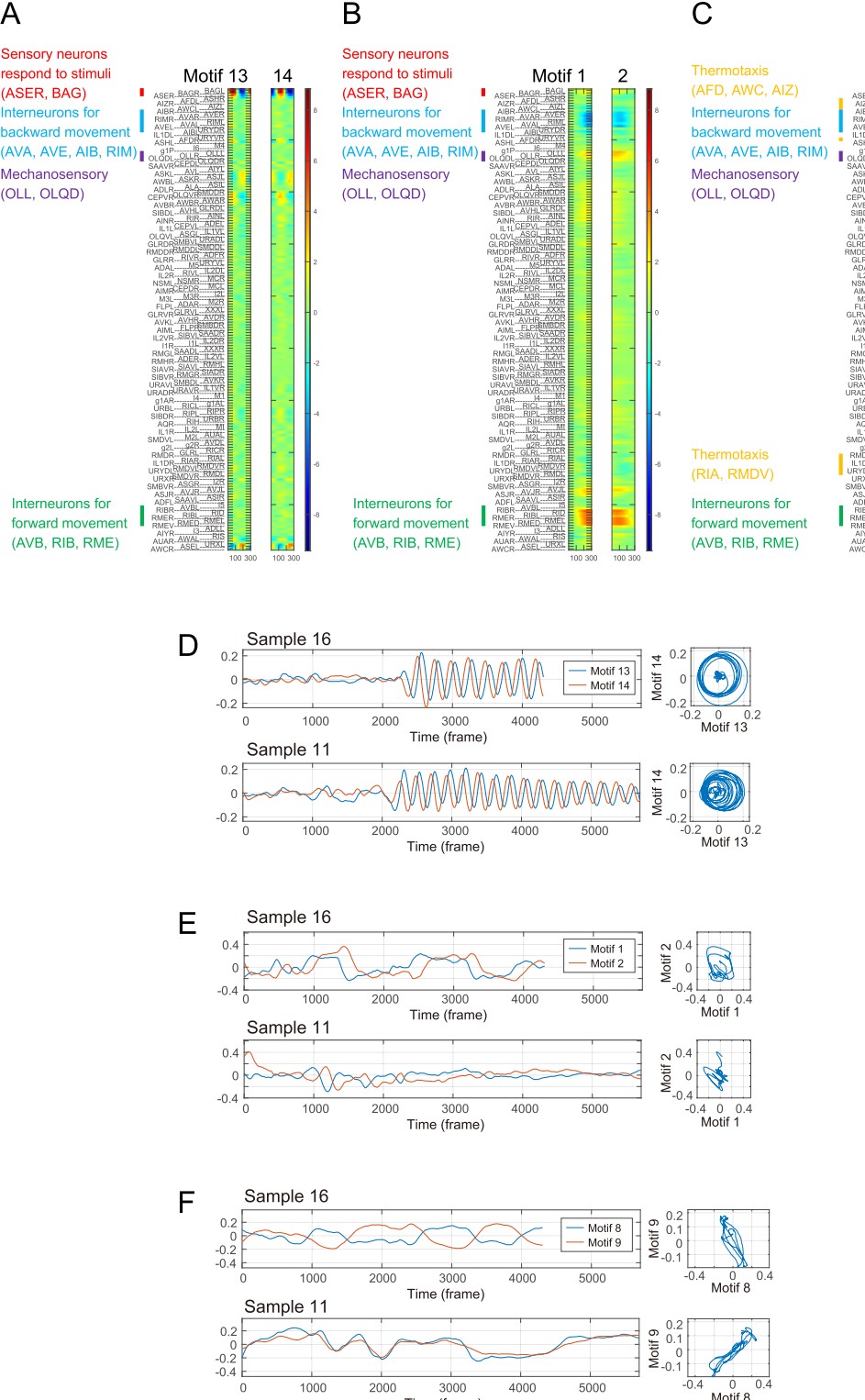

**Fig 5. Temporal motifs of neural activities obtained by TDE-RICA by matrix factorization.** (A) Motifs corresponding to sensory responses. (B) Motifs corresponding to forward and backward movements. (C) Motifs corresponding to the thermotaxis circuit. (D-F) Common features and individual differences of motif occurrences in the latent space. (D) Left: Occurrence of motif 13 (red) and motif 14 (blue) in sample 16 (upper) and sample 11 (lower). Right: Phase diagram of motif 13 and motif 14 in sample 16 (upper) and sample 11 (lower). (E) Left: Occurrence of motif 1 (red) and motif 2 (blue) in the same samples as D. Right: Phase

diagram of motif 1 (red) and motif 2 (blue) in the same samples as D. (F) Left: Occurrence of motif 8 (red) and motif 9 (blue) in the same samples as D. Right: Phase diagram of motif 8 (red) and motif 9 (blue) in the same samples as D.

ahead of time ($y_{i,j}(t+\Delta t)$) using the past time series of the presynaptic neurons (Fig 6A, see Methods for details). The target neuron's own previous activity was also included in the explanatory variables.

To reveal "important neurons and time points", we initially applied a dimension reduction technique called gradient kernel dimension reduction (gKDR) [32,33]. gKDR determines the $K$-dimensional subspace ($R^K$) of the space spanned by the explanatory variables (activity patterns of presynaptic neurons). The subspace is defined by a dimension reduction matrix $B_{i,j}$, such that $U_{i,j}(t) = \underline{X}_{i,j}(t)B_{i,j}$, where $\underline{X}_{i,j}(t)$ represents time-delay embedded activities of explanatory neurons (see Methods). gKDR aims to find $B_{i,j}$ such that $U_{i,j}(t) \in R^K$ contains sufficient information for predicting $y_{i,j}(t+\Delta t)$ (see Methods for details). An important characteristic of gKDR is that the estimation of $B_{i,j}$ can be achieved by solving a convex optimization problem with a simple eigenvalue calculation. Therefore, once $K$ and other hyperparameters are determined, the solution is uniquely determined for a given data set, eliminating the risk of getting trapped at ill-conditioned local minima.

As described in Fig 2, *C. elegans'* nervous system comprises several groups of neurons that exhibit synchronized activities and positive cross-correlations among their members. However, most of the activity changes of these groups are irregular and non-periodic, except for responses to regular sensory stimuli that were applied. This characteristic of the neuronal ensemble corresponds to the stochastic nature of the animals' behaviors. Thus, while it is unpredictable "when" a group of neurons are activated, they do it "simultaneously" once they are activated, driving a robust behavior. To reproduce this pattern, we employed a probabilistic model called gKDR-GMM, in which the relationship between the $K$-dimensional explanatory variables obtained by gKDR and the target variable was described by a joint distribution represented as a Gaussian mixture model (GMM). Each target neuron $i$ in each sample $j$ had its GMM model ($GMM_{i,j}$). For simulation purposes, we determined the conditional distribution of $y_{i,j}(t+\Delta t)$ on $U_{i,j}(t)$, namely $P(y_{i,j}(t+\Delta t)|U_{i,j}(t))$, from the joint probability distribution $P(y_{i,j}(t+\Delta t), U_{i,j}(t))$ modeled in $GMM_{i,j}$ (Fig 6A).

The gKDR-GMM model, in fact, consists of a collection of these multiple models ($B_{i,j}$ and $GMM_{i,j}$, $i = 1, 2, \ldots, M_j$; $M_j$ depicts the number of observed neurons in sample $j$), each describing the synaptic input to target neuron $i$. The meta-model comprising $M_j$ models characterizes the dynamics of the entire nervous system of animal $j$, as it represents the activation rule of all neurons in the system. The model was then run for simulation purposes (herein called free-run simulation). Whole nervous system simulation was performed simply by iteratively predicting the activity of $y_{i,j}(t+\Delta t)$ from activity data of presynaptic neurons up to time $t$. This process was repeated for $i = 1$ to $M_j$ (across all target neurons) to generate $X^*_{i,j}(t + \Delta t)$ (* denotes estimated values) and the whole process was repeated to proceed through time ($t+\Delta t$, $t+2\Delta t$, ....). Optimal hyperparameters for the gKDR-GMM model were determined by a grid search (See Experimental Methods, S1 Text—File 10).

Fig 6B shows some simulation examples by the gKDR-GMM model; all results are presented in S1 Text—File 11A-C. Two independent simulation repeats are displayed, which led to different results in each trial, highlighting the probabilistic nature of the gKDR-GMM model. Nevertheless, the overall ensemble patterns were well reproduced across simulations. During the hyperparameter search, we noticed that when we selected presynaptic neurons based on direct chemical or electrical connections to the target neuron (called direct links), the reproduction of network dynamics did not perform very well. This was expected since

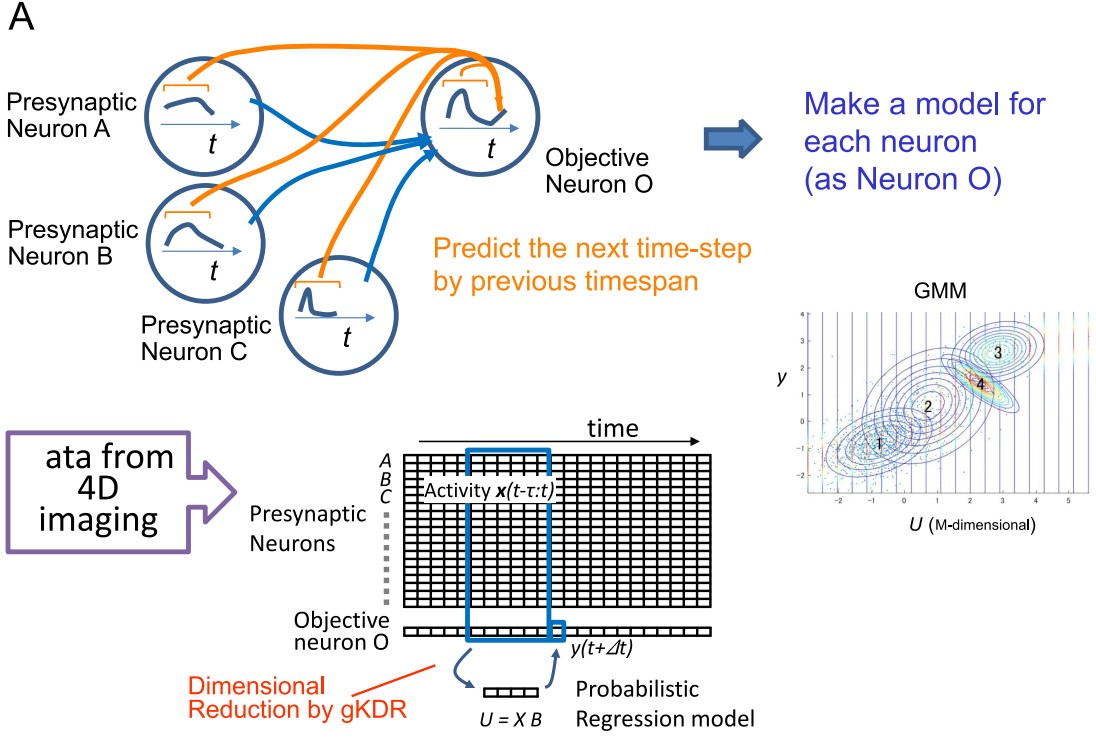

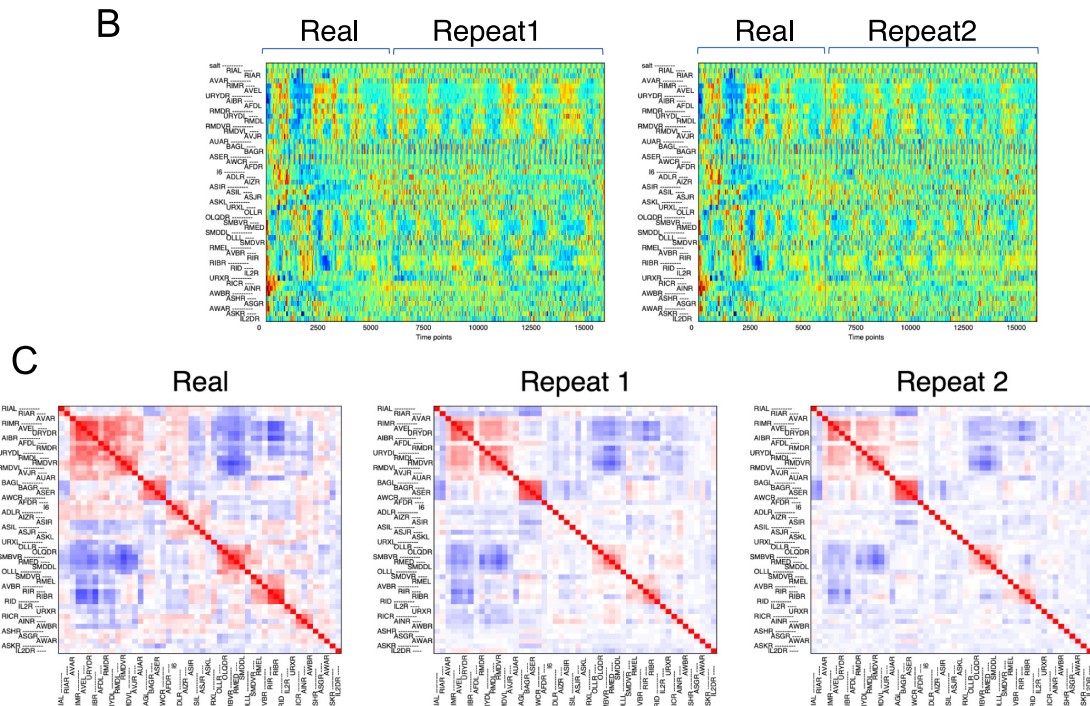

**Fig 6. Network modeling with gKDR-GMM.** (A) Overview of the gKDR-GMM model. The model learns to predict the target neuron activity (y(t+Δt)) from the previous activity of presynaptic neurons (X) (blue lines represent physical connections). To this end, gKDR reduces the dimension of presynaptic activities X and generates K-dimensional values U representing X, ensuring that sufficient information for predicting y is included in U. GMM models the joint probability of (U, y) as a weighted sum of Gaussian distributions fitted to real data. The conditional probability of y is determined from the GMM model and used for prediction. See Methods and S1

Text—File 23 for details. (B) Neural activities obtained from free run simulation by gKDR-GMM. The time span depicted as "real" is actual activity data, identical to Fig 2A whereas the time span depicted as "Repeat 1/2" displays simulation results. The results of two simulation runs are shown following the same order of neurons as Fig 2A. (C) Cross-correlation of neural activities. The left panel shows the cross-correlation of actual activities as in Fig 2B, shown for comparison. The middle and right panels are cross-correlations of the simulation results in (B). Red and blue color show positive and negative correlations, respectively. See S1 Text—File 11A-C for results of all samples.

approximately half of the neurons were missing from our data because of the difficulty in either observation or annotation (the maximum number of annotated non-random neurons was 110 out of ~190 head neurons). Consequently, we explored another option where missing presynaptic neurons were replaced by neurons with an additional step of synaptic connections, *i. e.*, presynaptic neurons for the missing neuron that was presynaptic to the target (called indirect links). Since this approach (indirect links) provided better results (S1 Text—File 10), we proceeded with the analysis based on the indirect links model, though our codes still support modeling with direct links.

Next, we assessed the model's reproduction capability in free-run simulations. While the results of free-run simulations may be different in each trial as described earlier, we anticipated that the relationships between real neurons would be preserved across simulation trials and that sensory information would be properly transmitted in the model nervous system. Therefore, we evaluated the cross-correlation of activity between neurons after free-run simulation by the model. An example of the correlation matrix is shown in Fig 6C and all results are presented in S1 Text—File 11A-C. To compare correlation matrices calculated from real activities and simulated activities, Fig 7A shows a scatter plot, real vs. simulated, of correlation coefficients between all pairs of neurons. Separate plots for each sample are presented in S1 Text—File 12. The correlations among groups of neurons observed in real data were largely preserved in the simulation, although the absolute value of correlation was lower in some neuron pairs. Considerable differences in the model performances between samples were also noted (Fig 7B).

Furthermore, we investigated the temporal relationships between neurons by evaluating the time-lagged cross-correlation of all pairwise combinations of neurons. Figs 7C and 7D, and S1 Text—Files 13 and 14 demonstrate that the optimal time-lags showing the largest correlation between each pair were maintained in the simulation for most samples (results of statistical tests are indicated in S1 Text—File 13). Further, we performed TDE-RICA using the same motif matrix to compare the dynamic relationship of major groups (S1 Text—File 15). The prominent relationship between motifs 13 and 14 was maintained in most samples and the observed behavior between motifs 8 and 9 shown in Fig 5F was also well-reproduced in the free-run simulation. Moreover, since the animals were stimulated by periodic changes in salt concentrations during 4D imaging, we assessed the periodic signals in each neuron by period average of the time series data and Fourier analysis (Fig 7E and 7F, and S1 Text—File 16). Fig 7E indicates that the periodicity of sensory stimulus is propagated over many neurons in the nervous system, as noted earlier (S1 Text—File 8); this characteristic was successfully reproduced in the free-run simulation by gKDR-GMM (Fig 7E and S1 Text—File 16).

These results demonstrate that our gKDR-GMM method effectively reproduces major neuron dynamics in a probabilistic manner. However, we note that this holds true for some but not all 4D imaging samples. Upon surveying all results, it appears that there must be distinct groups of correlated neurons for the dynamics to be accurately reproduced. In cases where most neurons exhibited globally correlated (epilepsy-like) activity changes (for example sample 6, 19, 20 in S1 Text—File 11B), the ensemble dynamics were difficult to reproduce. We speculate that such simultaneous activation of a vast majority of neurons may be caused by

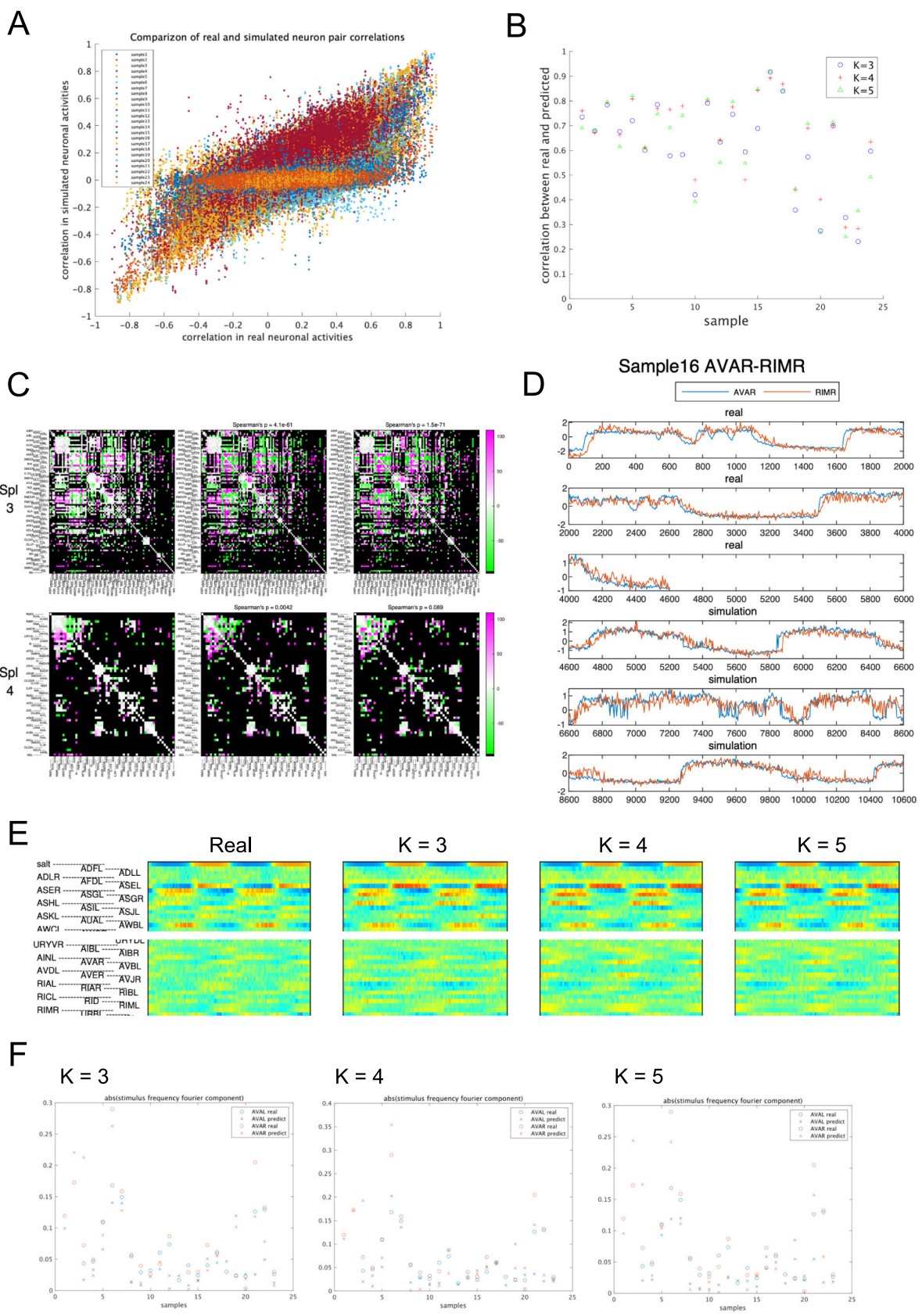

**Fig 7. Comparison of real and simulation results.** (A) For comparison of real data and simulated data, cross correlation between activities of neuron pairs are plotted for real (x axis) and simulated (y axis) activities for all pairs of neurons in each sample. Different samples are plotted in different colors. Plot for each animal can be seen in S1 Text—File 12. (B) Correlation coefficients of the real vs simulated relationship in (A) are plotted for each sample. Results of different models ($K = 3, 4, 5$) are also shown. (C) Lagged cross-correlation of all combinations of neurons. The lags with the best absolute cross-correlation are depicted color coded. Magenta and green indicate positive and negative lags, respectively. (D) Example of time series plot of a pair of neurons, AVA and RIM, showing a lagged correlation in most samples. (E) Periodicity of neural activities. Top row shows the salt stimulus (concentration range, 50–25 mM), which has a regular periodicity. Each of the real and simulation results as Fig 6B ($K = 3, 4, 5$ for simulation) was split into salt stimulus periods, overlaid and averaged to visualize the periodicity of the activity of each neuron. For visualization purposes, the period averages are shown repeated twice. (F) Fourier power of real AVAL/R activity (circles) and gKDR-GMM simulation results (crosses) with different model parameters A: $K = 3$, B: $K = 4$, C: $K = 5$. Only some of the samples show large periodic components.

extrasynaptic signaling, such as those by monoamine neurotransmitters or neuropeptides as observed recently [16].

## Noise-driven probabilistic behavior is essential for reproducing *C. elegans* brain activity

In the gKDR-GMM model, we intentionally incorporated the probabilistic nature of neural activity dynamics. However, it is important to compare the model behavior with the deterministic behavior of dynamic systems. This can be achieved by using a predicted time series of the conditional expectations $E(y_{i,j}(t+\Delta t)|U_{i,j}(t)))$, which we will refer to as the deterministic predictor.

Fig 8A and S1 Text—File 17A present an example of long-term free-run simulation with the deterministic predictor (middle row). In this case, the neuronal activities either became constant or changed slowly and did not mimic real activity profiles, and the correlation matrix did not resemble that of real activities (Fig 8B and S1 Text—File 17B). Removing the periodic sensory stimuli further degenerated the simulated activities, as it eliminated the activities synchronized to sensory input (middle row, right). Therefore, at least in our model settings, the deterministic predictors are unable to reproduce realistic activity patterns of the real neural network.

We hypothesized that the intrinsic noise in neurons might be a major source of the dynamic activity of interconnected neurons. To test this hypothesis, we introduced Gaussian white noise with constant size (*i. i. d.* noise) to the deterministic prediction. Surprisingly, this addition resulted in the emergence of stochastic synchronized activities, closely resembling those observed in real animals (Figs 8A and 8B, and S1 Text—Files 17A and 17B, bottom row). These results highlight the significance of considering the stochastic activities of each neuron when constructing a model to simulate the network properties of real animals.

## Robustness of the gKDR-GMM models and common characteristics of neural interactions

The gKDR-GMM model includes a considerable number of parameters ($K$ x presynaptic neuron number x embedding steps for $B_{i,j}$, and mean and covariance for each Gaussian component of $GMM_{i,j}$, for each neuron). Consequently, multiple sub-optimal solutions could be obtained for the time series data, leading to the question of which part of the model is reliable. To address this, we first evaluated the prediction capability of the model by conducting cross-validation. Time series data for each sample was split into three parts, and the gKDR-GMM model was trained with one of the three parts. The trained model was then used for prediction on the remaining two parts. Note that the cross validation will succeed only when neuronal interactions are stationary across sub-divisions of total recording time.

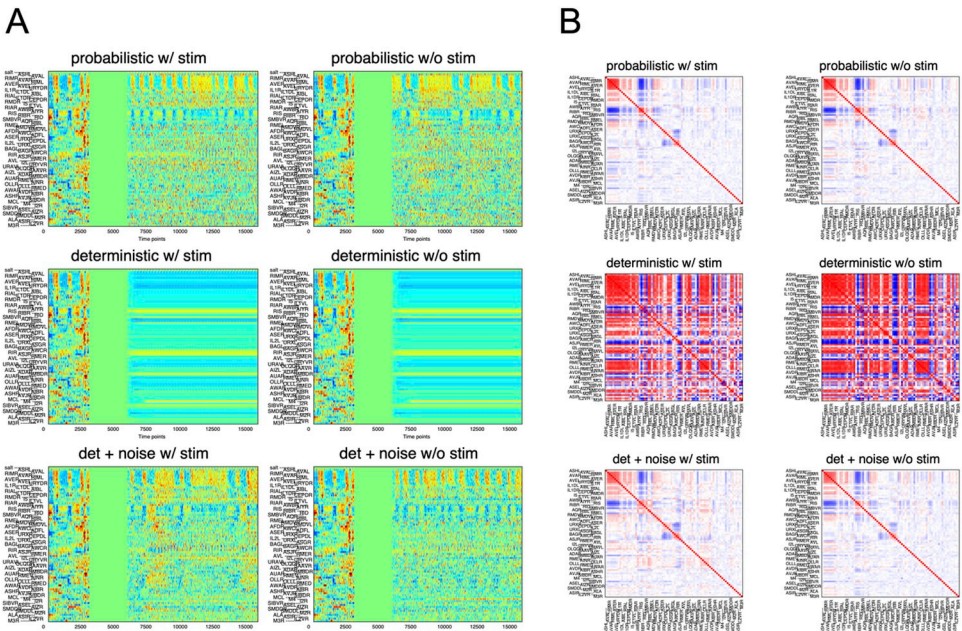

**Fig 8. Free-run simulation by deterministic prediction with or without added noise.** (A) The left side in each panel (0–1250 time points) shows the real data. The simulated results follow the zero-activity portion (1250–2500 time points) used as initial conditions for free run simulation. "w/ stim" indicates that periodic salt sensory input was added during the simulation while "w/o stim" indicates omission of the sensory input. "probabilistic" indicates regular gKDR-GMM (Fig 6) while "deterministic" represents prediction by expectation value (which is unique) for GMM. "det + noise" represents deterministic prediction with random independent noise added. (B) Cross correlation of the data obtained in (A). Color codes are as in Figs 2B and 6C.

Since gKDR-GMM is a probabilistic model, we scored the log-likelihood ($log(L)$) of the target neuron's real activity $y_{i,j}(t+\Delta t)$ (S1 Text—File 18A; all combinations of the training part and the testing part were evaluated). We then performed bootstrap tests to evaluate the contribution of presynaptic neurons to the prediction of target neuron activities. To do this, the presynaptic neuron data $\underline{X}_{i,j}$ were randomly permuted along the time axis, and prediction results were obtained using the permuted $\underline{X}_{i,j}$. The distribution of $log(L)$ for 100 repeated random permutation represents the null hypothesis, which assumes presynaptic neuron data do not lead to better prediction. S1 Text—Files 18B and 18C show that in more than half (58.6%) of all models, target neuron activity is significantly better predicted, indicating that gKDR-GMM properly models the mapping between the inputs from presynaptic neurons and the target neuron activities. Considering the possibility of non-stationarity of neuronal interactions across recording time, we can select and utilize one of the three models that were successful in the cross validation, which likely represents stationary interactions (see below).

How does the neural network integrity, particularly the correlation of activities among neurons, emerge from our model? Our gKDR-GMM model includes estimates of synaptic transmission's nature, representing a complex relationship between multiple presynaptic neurons at previous time points and the target neuron activity as a probabilistic distribution composed of multiple Gaussian distributions. To simplify the estimation of synaptic weights, we calculated the average of the gradient, $\partial E(y_i)/\partial x_h$. For this measure, a positive value indicates excitatory transmission, where the more active the $h$ input is, the more active the target neuron $i$ is expected to be at the next time step. Conversely, a negative gradient indicates the inhibitory transmission where the relationship is reversed. For clarity, we refer to this calculated gradient as the "synaptic weight".

In our gKDR modeling, we selected every five time points to reduce the data size (see Materials and Methods). As a result, we had a choice of five different phases of data point selection, starting from $t$ = 1, 2,, or 5. Due to significant noise in our data, these different phases could lead to considerably different models. To extract robust synaptic weights among the models, we tested gKDR-GMM models constructed with five different phases and three different $K$ values ($K$ = 3, 4, 5; $K$ being gKDR dimension) by cross validation, as mentioned above. We selected only the models that passed the cross validation test and attempted to extract consistent synaptic weights among the different models generated for the sample.

Fig 9A and S1 Text—File 19A display the synaptic weights estimated by all qualified models. The results demonstrate that for certain neuron pairs, the estimation is highly consistent across different models, suggesting that these synaptic weights need to be non-zero to be consistent with the observed neuronal activities. Based on these observations, we estimated the statistical consistency across models as shown in Fig 9B and S1 Text—File 19B. In these figures dark red and blue indicate strongly consistent positive and negative synaptic weights, respectively. Members of the same class, such as left and right members (designated as XXXL and XXXR, XXX being a class name) often showed similar weights, consistent with overall left-right symmetry of the neurons and their interconnections.

We further aimed to extract synaptic communications that are conserved among different samples. In this direction, Fig 9C and S1 Text—File 19D present all estimated synaptic weights from each tripartite time span of each sample. Similar to Fig 9B above, Fig 9D and S1 Text—File 19E assess the consistency of synaptic weights, this time across samples.

One prominent feature of the whole brain dynamics is the presence of backward (group A) and forward (group B) groups, and how these correlated/anticorrelated activities are generated is an important and open question, with only reciprocal inhibition so far suggested [29]. By close examination of the results in Fig 9A–9D (and S1 Text—Files 19A, 19B, 19D, and 19E; also Fig 10A described later), it is found that there are highly consistent networks of positive synaptic communications among AIB, AVA, AVE and RIM classes of neurons. In terms of the group B neurons, communications are less consistent, which may correspond to the different sizes of the correlated clusters among different samples (Fig 2B). Among these, RIB had consistent negative synaptic weights to AVA and AVE. The opposite communications were less prominent, but strong in some samples. None of other B type neurons such as AVB, RID and RIS showed consistent weights across samples, but can have strong communications with A type neurons in some samples. Also, RMD, head motor neurons, and URY, non-ciliated sensory neurons receive consistent positive inputs from the A type neurons.

As described earlier, the gKDR-GMM models are based on time-delay embedding, which means that the gradient $\partial E(y_i)/\partial x_j$ representing synaptic weight has a time-lag dimension, which was not fully described in the previous section. The lag dependency of synaptic weights was graphically presented in S1 Text—Files 19C and 19F. For neurons that exhibited sensory-related activities, the weights reflected the periodicity of sensory stimuli applied during the experiments. In many other neurons, the weight rapidly decayed to zero at a short time lag, indicating that the presynaptic activity immediately preceding time $t$ (lag 0) affected the activity of the postsynaptic target neuron at time $t+\Delta t$. There were, however, also neuron pairs in which the decay is slower or even the weight remained either positive or negative throughout the whole span of the lag. A prominent example is the neurons in the core backward command circuit (group A). Synaptic weights between AVA and RIM showed a fast decay, while AIB to RIM weight was more extended (Fig 9H), suggesting a longer decay time in the synaptic communication. This is consistent with the prior knowledge that AIB conveys sensory input to slowly bias the probability of reversal [34].

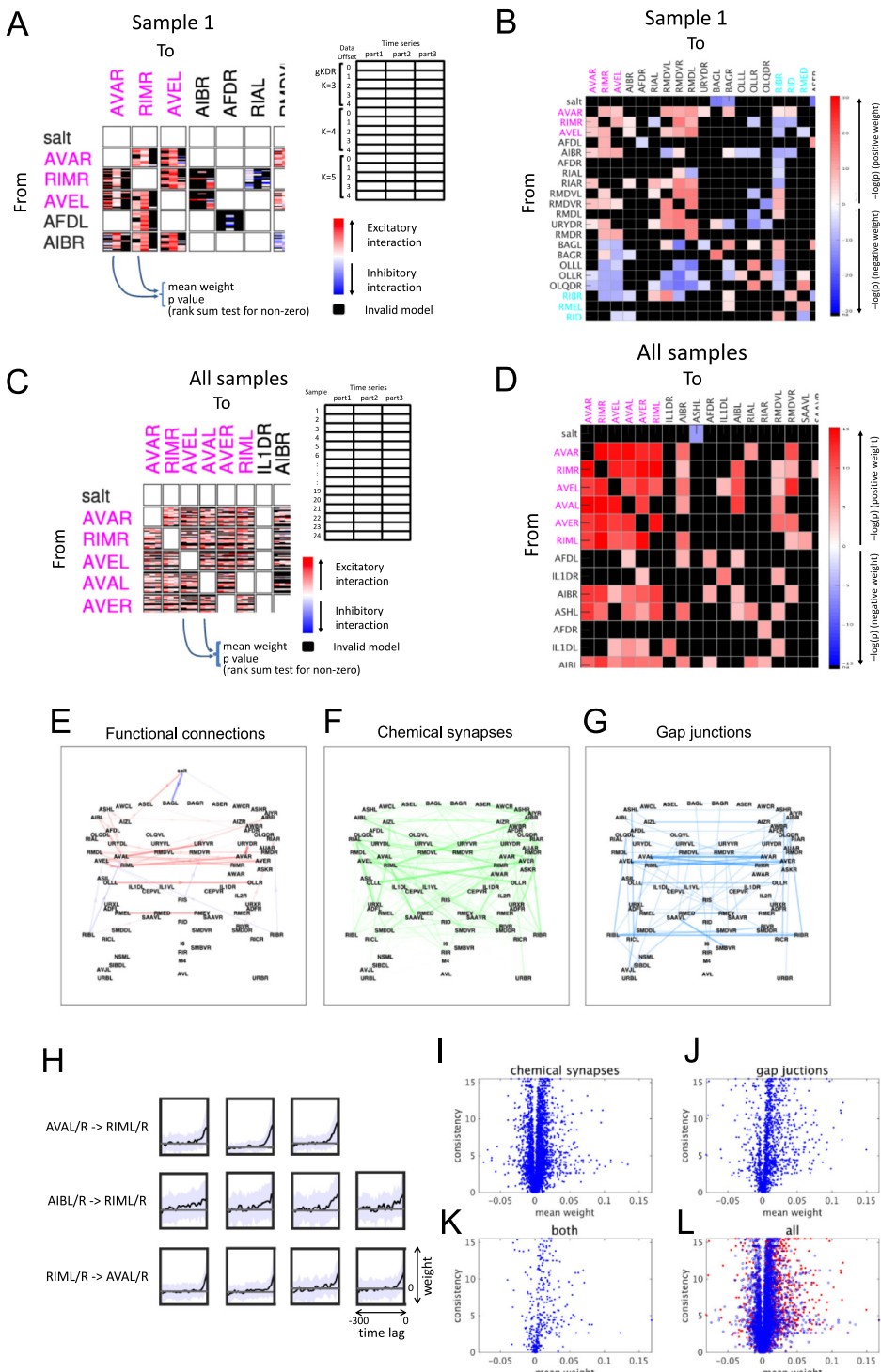

**Fig 9. Estimation of synaptic weights from gKDR-GMM models.** (A) Estimated synaptic weights from different models. Each box shows model-estimated synaptic weights from the neuron on the *y* axis to the neuron on the *x* axis. In each box, model-estimated weights are shown in 15 x 3 cells as summarized on the right, where 15 rows are results from models using different *K* for gKDR using offset 0–4 of the data, and 3 columns show results from three parts of split time series. Models that did not show significance in bootstrap cross validation tests at p < 0.01 are filled in black. (B) Consistency across 15 models in each box in (A) were tested by Wilcoxon's rank-sum test and p values are shown in color codes. Darkness of the color shows –log$_{10}$(p), while red colors show positive mean weights and blue colors show negative mean weights. (A) and (B) show part of the table for sample 1 as examples. Full figures are shown in S1

Text—Files 19A and 19B. (C) Estimated synaptic weights from different samples. In this figure, estimated weights from 15 models in each box of (A) are averaged, and shown in 24 x 3 arrangement, as depicted on the right. (D) Consistency across 24 samples in each box in (C) were tested by Wilcoxon's rank-sum test and p values are shown in color codes as in (B). Darkness of the color shows $-\log_{10}(p)$, while red colors show positive mean weights and blue colors show negative mean weights. (C) and (D) show part of the table as examples. Full figures are shown in S1 Text—Files 19D and 19E. In (A)-(D), the order of neurons is the same as that in Fig 2C. Group A and B neurons are shown in magenta and cyan, respectively. (E) Mean synaptic weights from all samples are shown as graph representation. Only the weights that showed consistency at FDR < 0.005 in (D) among direct synaptic connections are shown. (F, G) Graph representation of chemical synapses (F) and gap junctions (G) are shown between neurons shown in (E). (H) Mean (line) and standard deviation (light blue shade, across samples) of estimated synaptic weights at each lag between the neuron classes indicated. (I-L) Mean synaptic weights vs. consistency (-log(p) in (D)) were plotted for each of chemical synapses and gap junctions; for pairs of neurons with only chemical (I), only electrical (J) or both synapses (K), and for all pairs of neurons (L). In (L), gap junctions are shown in red dots and chemical synapses are shown in blue circles.

Having obtained an estimate of functional connectivity, we examined the contribution of gap junctions and chemical synapses to the communications. As depicted in Fig 9E–9G, the functional network exhibited greater similarity to the gap junction network. To provide a more quantitative analysis, Fig 9I–9L display the estimated consistency and mean synaptic weights for each of the chemical and electrical synaptic contacts. These figures clearly indicate that gap junctions tend to generate stronger and more consistent positive connection weights, whereas chemical synapses contribute to weaker but consistent positive and negative synaptic weights. Thus, while previous electron microscopy reconstruction identified the structural connectome in this organism, our model advanced further and estimated functional synaptic communications based on the whole-brain activity imaging data.

As described earlier, we applied periodic sensory stimuli of salt concentration changes to the animals, and the activity of downstream neurons, including those of the group A and B command interneurons, included weak periodic components corresponding to the sensory stimuli (S1 Text—File 8). These periodic components were often reproduced by the gKDR-GMM models (Fig 7E and S1 Text—File 16). To see how sensory signal is transmitted to command interneurons, synaptic weights between these neurons, as well as primary interneurons that receive direct synaptic inputs from sensory neurons, were estimated. In Fig 10A, red/blue colors indicate the mean synaptic weight, while green color indicate coefficient of variation (CV) of synaptic weights among samples. Therefore, pure red and blue indicate consistently positive and negative connections, respectively, while green color indicates connections that switch between positive and negative in different animals. There are consistent positive weights among group A neurons, and consistent negative inputs from RIB are received by these neurons, as described earlier. On the other hand, communication from salt sensing neurons to command neurons, as well as primary interneurons are highly variable (S1 Text—File 20). These results strongly indicate that sensory information is transmitted in a variety of routes among different samples, which may reflect behavioral variability and plasticity.

## Virtual optogenetics to assess propagation of neural activities through the network

Another application of the gKDR-GMM model is virtual optogenetics. Because transmission of neural activities is modeled by gKDR-GMM, we can artificially activate a selected neuron and simulate the activity of all other neurons thereafter. For this experiment, we only used the data from samples 13 to 24, in which salt stimuli were not applied in the first half of the recording (unstimulated period). Because the gKDR-GMM model predicts neural activity based on previous activities of presynaptic neurons, the simulated activity profiles depend on the initial conditions. In this test, we used the unstimulated period of actual recordings as initial conditions to mimic actual optogenetics experiments in real animals.

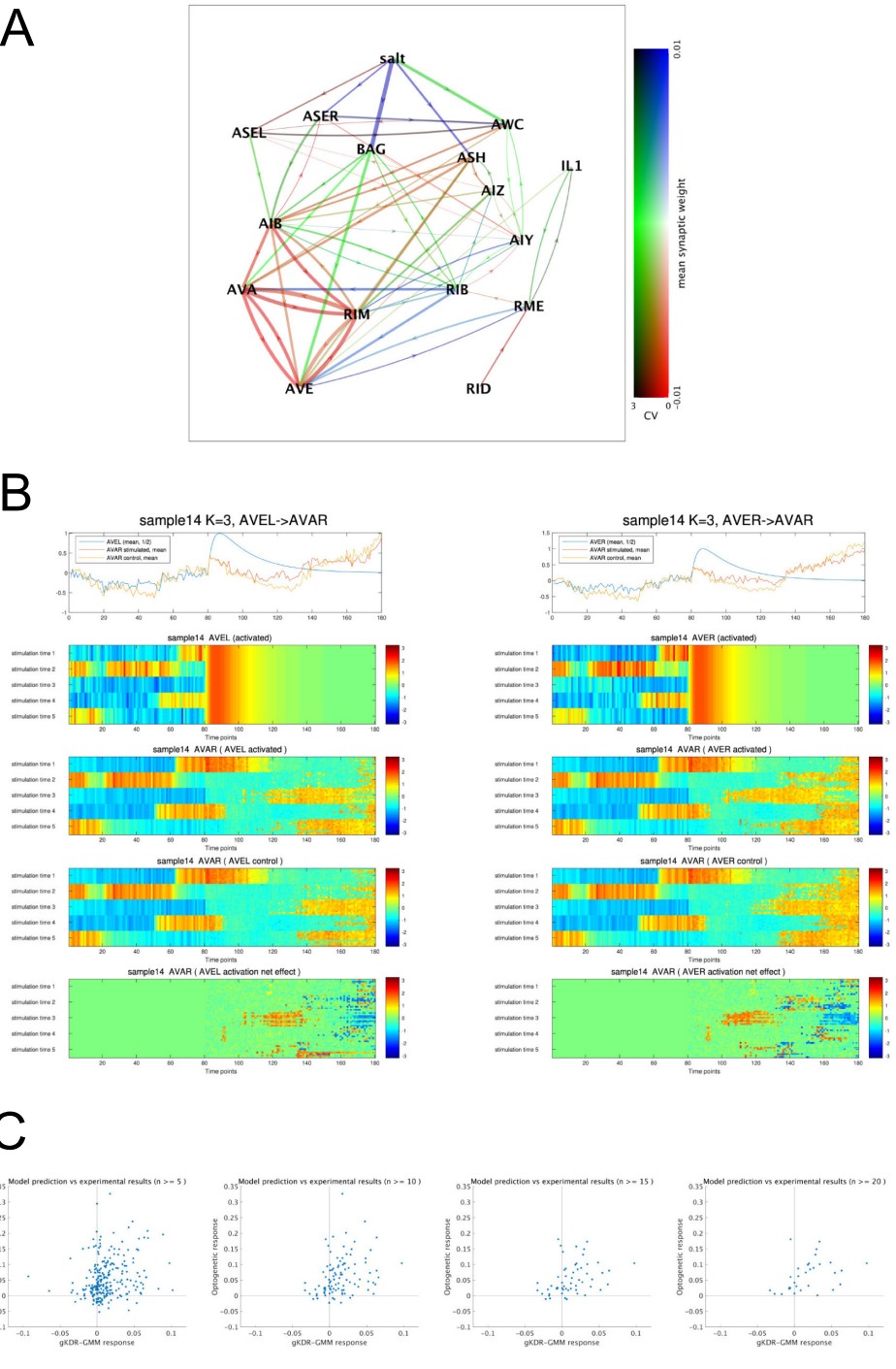

**Fig 10. Estimation of signal flow in the neural network.** (A) Estimation of synaptic weights across samples. Salt sensory neuron classes ASE, BAG, AWC, and ASH, primary interneurons and command interneurons of groups A and B as well as some other neurons are shown. Blue and red colors indicate mean synaptic weights (positive and negative), green color indicate coefficient of variation. Thickness of the connections reflect numbers of samples contributing to each estimation. Only direct synaptic connections are shown. (B) Examples of virtual optogenetics. Simulation was performed using the gKDR-GMM model obtained from the neural activity data of sample 14, with a hyperparameter $K = 3$ in this example. AVEL neuron (left) or AVER neuron (right) was artificially activated at five three different time points (depicted stimulation time 1 to 5) in the real activity time course of sample 14, which were aligned at time point 80 in the x axis. Real imaging data are shown for time points 0–80, and using them as initial

values, simulation was started at time point 80 with all neurons observed in sample 14. Simulation was performed 10 times each, and resulting activity of neurons AVEL/AVER (row 2) and AVAR (rows 3 and 4) are shown in pseudocolor. In row 4, AVEL/AVER was not stimulated but instead set to zero, as a control. Top row shows the average of all results in each of rows 2–4. The difference between activated and control is shown in row 5. Results of all neuron pairs are shown summarized in S1 Text—Files 21A and 21B. (C) Our results of virtual optogenetics simulation as shown in (B) were compared with the real optogenetic experiments reported by [35]. Because of the high degree of variability of the network states and its effect on signal propagation as shown in (B), only results with sufficient numbers of experiments were compared. Considering the sample-to-sample variability in our gKDR-GMM model, only neuron pairs that were observed in at least eight samples were included. For the data from [35], only neuron pairs that were tested at least 5, 10, 15 and 20 times, as depicted above each panel, were included. For neuron pairs that satisfy both of the above, extent of activation or inactivation of the observed neuron after activation of the stimulated neuron was plotted for our results (*x* axis) vs by [35] (*y* axis). See Materials and Methods for quantification.

In the example shown in Fig 10B, AVEL or AVER neuron was activated at five different time points in the unstimulated period of actual recording of sample 14, and the effect on AVAR activity is depicted. As a control, the same neurons were clamped at activity 0 ("AVEL/R control"). Each test was repeated 10 times. As previously shown in Fig 2 and S1 Text—File 1, there is a brain-wide spontaneous activity even without applied sensory stimulations. We therefore subtracted each neuron's control activity from that after optogenetic activation, to reveal the response of each neuron to the artificial activation (Fig 10B, bottom). The effect considerably differed depending on stimulation time, strongly arguing that network states affect signal propagation. For example, in Fig 10B stimulation time 3 and 5, AVAR is inactive before stimulation and spontaneously activated thereafter, and AVE stimulation accelerates the activation. At stimulation time 4, AVE stimulation temporarily delays spontaneous inactivation, while at other time points activation of AVE has no systematic effect. Quantification results of these experiments are provided in S1 Text—Files 21A and 21B.

Randi et al. [35] recently reported systematic optogenetic stimulation and whole-brain calcium imaging experiments to assess signal propagation in the *C. elegans* nervous system, which can be considered the real-world counterpart of our *in silico* experiments in Fig 10B and S1 Text—File 21. Unlike modeling approaches, however, a control experiment cannot be paired with each experiment, because network states continuously change and it is impossible to observe experimental response and control response at the same time. Randi et al. therefore averaged the results and employed statistical approaches (Fig 2A in [35]). We note that overall, negative responses are rare in their results; the reason for which is unknown. For the reasons above, to compare our simulation results with their experimental results, we focused on neuron pairs to which sufficient numbers of trials were performed in both studies and compared the mean responses of each set of results. We took only neuron pairs for which we have data from eight or more samples (two thirds of all). Among these pairs, we further selected neuron pairs for which Randi et al. performed at least 10, 15, 20 or 25 tests. The Pearson's correlation coefficients of mean responses in the two experiments were 0.250, 0.342, 0.433 and 0.486, respectively, for neuron pairs selected by these criteria (Fig 10C, n = 246, 100, 56 and 31 neuron pairs, respectively). Therefore, results of the two approaches are correlated positively provided that sufficient numbers of experimental results were obtained. We therefore conclude that our gKDR-GMM modeling based solely on whole-brain calcium imaging observations can be used for estimating the signal propagation in the network.

## Discussion

In this study, we performed calcium imaging of the *C. elegans* head region using the calcium probe Yellow Cameleon 2.60 expressed in all neuronal nuclei. Through 3D segmentation, tracking and annotation, we successfully obtained a comprehensive "whole-brain" neural

activity dataset comprising 24 samples, which included neural activity data with neuronal identity labels. To the best of our knowledge, this represents the most extensive annotated neural activity dataset for *C. elegans* hermaphrodites reported to date. For the analysis of this dataset, we developed and applied two analytical methods, namely, TDE-RICA and gKDR-GMM. TDE-RICA demonstrated its effectiveness in decomposing network dynamics into motifs involving specific subsets of neurons, thereby facilitating a more profound functional dissection of neural activity patterns. On the other hand, gKDR-GMM allowed us to estimate synaptic strength and time constants, which provided a model for understanding how core networks generate correlated activities and are regulated by sensory inputs. Moreover, gKDR-GMM revealed the critical role of noise in realistic neural networks and enabled realistic simulation of whole-brain activities including virtual optogenetics experiments.

Although our recording was performed in animals restrained in the same chamber and similarly stimulated by the chemoattractant NaCl, there were considerable differences in activity patterns among individual samples. We currently do not know whether these differences stem from individual variations in connectome structure that occur during embryonic or post-embryonic development, as reported in previous studies [9], or they are due to different prior experiences resulting from subtle variations in environmental stimuli. Alternatively, the difference may not reflect individual variations but might be a consequence of fluctuation over time of the network dynamics, similar to dwelling and roaming observed in the locomotion behavior of *C. elegans* [16,36]. In addition, a part of the difference could be attributed to the fact that only a fraction of all neurons was annotated in our dataset, though obvious differences exist even among commonly observed neurons.

To overcome the limitations of aforementioned partial annotation and inter-sample variability, and extract the activity profiles common among different samples, we employed independent component analysis (ICA). ICA is commonly used for signal separation, but when combined with time-delayed embedding, it allowed us to extract common dynamic patterns from the entire brain network. Additionally, by utilizing matrix factorization, we could compensate for the missing data and extract global patterns. The results revealed multiple sets of neurons engaged in major dynamic patterns, including sensory response dynamics (motifs 13 and 14) and spontaneous motor dynamics (motifs 1 and 2). Although the TDE-RICA method successfully extracted stimulus-dependent and spontaneous motor signals as separate components, it could not clarify how stimulus-dependent signal affects motor signals. This coarse-grained approach might be more suitable for examining global dynamics across all individuals, rather than analyzing subtle stimulus-dependent signals.

We then estimated the synaptic weights along the physical network structure of *C. elegans'* nervous system, where we aimed to construct data-driven phenomenological models rather than physical models. Instead of employing a differential equation approach, which requires assuming specific transmission functions, we utilized time-delay embedding and nonlinear regression models. Using nonlinear models is crucial for reproducing neuronal events, since synaptic transmission is intrinsically nonlinear. Given that there are on average 10–20 presynaptic neurons (S1 Text—File 22), the dimensionality of the explanatory variables is quite large. Therefore, selecting a proper dimensionality reduction method is crucial. Commonly used dimensionality reduction methods such as PCA [10], ICA, kernel PCA, t-SNE [37,38], UMAP, etc. fail to represent the relationship between explanatory variables (activity of presynaptic neurons in our case) and response variables (postsynaptic neuron activity ahead of time). This could result in removal or underweighting of variables (neurons and time points) that strongly influence the response of the target neuron. Therefore, we preferred dimensionality reduction that considers the response. Examples of linear approaches for this purpose are canonical correlation analysis (CCA) [39] and partial least square (PLS) [40,41] among others. These

methods seek a low dimensional projection ($U$) of explanatory variables ($X$), so that $U$ has a high correlation coefficient or high covariance with the response variable, $y$. However, as mentioned earlier, the relationship between $X$ and $y$ is likely nonlinear. Hence we chose kernel dimensionality reduction (KDR), which is based on the kernel method and does not assume linearity.

Through gKDR-based dimensionality reduction and probabilistic regression by GMM, we were able to reasonably reproduce the overall dynamics of the *C. elegans* brain. However, we need to be cautious that the set of synaptic weights is not necessarily the only solution that reflects the actual synaptic weights in the real animals. To address this concern, we evaluated multiple models generated from different time spans of the data with different time point selections, as well as different values of the modeling parameter $K$ (dimension reduction size). This comprehensive approach allowed us to differentiate between robust and variable aspects of the results, with the former providing valuable insights into how the network dynamics is shaped.

A remarkable finding from our modeling approach is that a probabilistic model is essential for reproducing realistic network activity of *C. elegans*, exemplified by the coordinated but stochastic activation-inactivation of command neurons. A deterministic model obtained by the same gKDR approach generated only decaying activities. This observation and further simulation led us to propose that noise, or time-independent stochastic activities, intrinsic to the nervous system is driving the network dynamics. Electrophysiologists often observe spontaneous and irregular activities of neurons. Even in *C. elegans* which does not show $Na^+$-based action potentials, stochastic activities are often observed, especially in interneurons (for example [42]). Irregular activities are observed even in isolated neurons ([43]). Thus such kinds of small irregularities in multiple neurons likely sum up to cause stochastic state transitions in the neural network dynamics.

The advantage of the synapse-based model is its ability to identify crucial neurons and synapses that are important for specific aspects of the whole-brain dynamics. Our analysis of the estimated synaptic weights suggested a general rule that the correlation of the activity in a pair of neurons is not necessarily formed by the neurons themselves, but each neuron is influenced by inputs from a group of neurons, which cause correlated activities. Because we restrained *C. elegans* in a microfluidic chip and applied sensory stimuli, naturalistic behaviors cannot be observed. However, neuronal commands such as forward-backward commands can be read out from neural activities. In natural environments, *C. elegans* shows chemotaxis to various chemicals including salt. Chemotaxis is achieved by several types of behavioral responses, including klinokinesis. In klinokinesis, probability of reversal is increased for around ten seconds after a salt concentration decrease [34], which contributes to gradual migration towards salt, because migration away from the salt leads to increased reversal followed by turning. Our 4D imaging results indicated that while backward neurons (group A neurons) are spontaneously activated in a coordinated fashion, it shows a weak correlation with sensory inputs, which is consistent with previous observations [34]. Our analysis of periodicity (Fig 4B and S1 Text—File 8) and synaptic weight prediction (Fig 10A, and S1 Text—Files 19 and 20) provide insights into how sensory stimulus is transmitted to backward command neurons: sensory transmission operates as a distributed system, where information from sensory neurons is transmitted not through a single specific path, but through multiple paths, which collectively drives the sensory-driven control of behavior. In addition, the paths are variable between individual animals (Fig 10A and S1 Text—File 20), which is also a remarkable finding obtained in this study.

For the core neural network of *C. elegans*, group A neurons governing backward movement and group B neurons governing forward movement, our estimation of synaptic contact

revealed that almost all-to-all communications between AIB, AVA, AVE and RIM neurons drive the group A dynamics, while RIB neuron plays a key role in generating anti-correlation between groups A and B. RIB is known to be GABAergic and assumed to be inhibitory, and while it is likely that RIB is in turn negatively regulated by group A neurons, further investigation is needed to clarify this pathway.

In the analysis of massive brain activity data, various analytical methods are utilized such as cross correlation, as a measure of functional connectivity between brain areas [44,45], and Granger causality [46]. However, these methods are merely descriptive of the relationships between neurons or brain areas, and their results cannot be utilized for long-term prediction or simulation. Therefore, there is a pressing need for new methods that incorporate causal structures of the data and offer predictive capacity. Our approach exemplifies one such endeavor. This methodology could be employed for large brains, for which connectome data is increasingly being obtained [2] as well as increasingly detailed activity data from wide-field calcium imaging, fMRI, EEG and MEG [4]. While our current analyses were applied to connectome data of *C. elegans*, the same method could potentially be adapted to analyze neuronal activity data in cases where no connectome information is available, by assuming all-to-all connections.

## Materials and methods

### Strains and culture

Animals were raised at 20˚C under standard conditions on nematode growth medium (NGM) plates with *E. coli* OP50. For 4D imaging, the following strain was used [23]: JN3038 *qjIs11[glr-1p::svnls2::TagBFPsyn, ser-2(prom2)p::svnls2::TagBFPsyn]; peIs3042[eat-4p::svnls2::TagRFP675-syn, lin-44p::GFP]; peIs2100[H20p::nls4::mCherry]; qjIs14[H20p::nls::YC2.60]*. Transgenic strains were generated by germ-line transformations in which we co-injected the gene of interest with a visible transformation marker (*lin-44p::gfp* etc.) into the animals. The transgene was then integrated into a chromosome by UV irradiation.

### Microscopic setup

The microscope system was developed as previously described [23]. Briefly, the system consists of a spinning-disk confocal microscope and three cameras (one CMOS camera and two EM-CCD cameras). A piezo actuator is attached to the objective lens of the confocal microscope to enable high-speed 3D imaging. The CMOS camera is used to capture images for cell detection and tracking, and the two EM-CCD cameras are used to measure neuronal activity (Yellow Cameleon imaging).

### Image acquisition

The detailed microscope settings for imaging were previously described [23]. Animals were raised on standard NGM plates until young adults, and further incubated overnight on pre-imaging NGM plates with 50 mM of NaCl. Osmolarity of pre-imaging plates was adjusted to 350 mOsm with glycerol. Animals were introduced into a PDMS microfluidic chamber and stimulated with a change in salt concentration from 50 mM to 25 mM or from 25 mM to 50 mM every 30 seconds (the period was 60 seconds). We used a modified version of the olfactory chip [19] for the microfluidic device. The stimuli were delivered to the animals by switching the imaging solutions (25 mM potassium phosphate (pH 6.0), 1 mM $CaCl_2$, 1 mM $MgSO_4$, 0.02% gelatin, NaCl at the indicated concentration and glycerol to adjust their osmolarity to 350 mOsm).

For the 3D images for neuronal annotation (called annotation movie), 50 slices per volume were taken to cover the entire body of animals (0.72–1.00 μm / slice). The size of each image is 1024*150 pixels. A total of 8 volumes were taken, and the best one was used for neuronal annotation. To acquire images for measuring neural activity (called activity movie), 22 slices per volume were taken for the same area (1.62–2.29 μm / slice). The size of each image is 256*64 pixels. A total of 6,000 volumes were taken at a rate of about 4 volumes per second (about 25 minutes in total).

## Cell detection, annotation and tracking

All the nuclei in a volume in the 3D images for neuronal annotation (called the annotation movie) were detected by our image analysis pipeline roiedit3D [21] and corrected manually. We detected 201.9 ± 15.5 (mean ± standard deviation) nuclei on average from 24 samples.

The cells were annotated with neuronal identity based on the expression patterns of cell-specific promoters as previously described [23]. We annotated 146.8 ± 23.1 (mean ± standard deviation) cells on average.

For cell tracking, our cell tracking pipeline CAT was used. CAT selects the volume most similar to the volume in which the cells were identified from the activity movie. The volume in the annotation movie was registered to the volume in the activity movie by B-spline transform implemented in elastix [47]. The ROI information including position, size, and annotation of nuclei was copied from the annotation movie to the activity movie. CAT also constructs a shortest path tree, where the volumes of the activity movie correspond to the nodes of the tree, and the similarities between the volumes correspond to the edges of the tree. The volumes connected by the edge were registered by the B-spline transform, and ROI information was copied. CAT run on the Shirokane3 and Shriokane5 supercomputing systems of the Human Genome Center (the University of Tokyo), and processed an activity movie within about 8 hr. The tracking results were checked and corrected manually by using roiedit3d. The fluorescence intensities of CFP, YFP, and mCherry were obtained by least square fitting of the tri-variate Gaussian mixtures, which correspond to the tracked ROIs, to the nuclei in the volume [21]. If the worm moved outside the field of view, the error periods were removed from the dataset.

## Pre-treatment

To remove noise, a median filter with a 5-point window was applied to the time series of CFP and YFP intensities. If the filtered CFP intensity is smaller than 1/10 of the median of the filtered time series of CFP, the time point was regarded as an outlier. CFP and YFP intensities at the outliers were set as NaN (missing), and were completed by a median filter with a 4-point window. The missing values can be completed if the missing values are sparsely distributed. The outlier neurons were removed from the dataset if the number of time points of the outlier is larger than 400 or any missing values remained in the completed time series. We also removed non-neuronal cells including hypodermal cells. The ratio of YFP over CFP was calculated, and the linear trend of the ratio was removed to compensate for the photobleaching. The time series of the ratio was subtracted by its mean and divided by its standard deviation for scaling. The scaled ratio of YFP over CFP was regarded as the neural activity. The obtained whole-brain activity dataset contains the neural activity of 139.6 ± 24.9 (mean ± standard deviation) cells on average from 24 samples, and covers a total of 177 out of 196 cells in the head region of animals. For gKDR, noisy neurons were removed from the analyses. Specifically, only neurons with autocorrelation of greater than 0.3 at a lag of 20 time points were included and those that did not meet this criteria were considered missing.

## Time-delay embedding and Reconstruction ICA (TDE-RICA)

Let $x_{i,j}(t)$ be an activity of neuron $i$ (from 1 to 177) of sample $j$ (from 1 to 24) at time point $t$ (from 1 to 6000). The embedding of neuron $i$ in sample $j$ at time point $t$, $X_{i,j}(t)$, is $X_{i,j}(t) = [x_{i,j}(t-(k-1)\tau), x_{i,j}(t-(k-2)\tau), \ldots, x_{i,j}(t)] \in R^{1 \times k}$, where $k = 300$ and $\tau = 1$. We choose these values arbitrarily so that the motifs obtained by the following analysis can capture the large-scale dynamics of neural activities. The embedded time series of neuron $i$ in sample $j$ is
$X_{i,j} = [X_{i,j}^{\top}((k-1)\tau + 1), X_{i,j}^{\top}((k-1)\tau + 2), \ldots, X_{i,j}^{\top}(6000)] \in R^{300 \times 5701}$.

We used Reconstruction ICA [28] implemented in Matlab so that the captured components and weights can reproduce the original data. Because the RICA cannot handle missing values, 94 neurons of 10 samples (with no missing values) were selected from the whole dataset of 177 neurons of 24 samples (with missing values).

The embedded time series of the 94 selected neurons in sample $j$ is $X_j = [X_{1,j}^{\top}, X_{2,j}^{\top}, \ldots, X_{94,j}^{\top}]^{\top} \in R^{(94 \times 300) \times 5701}$, and the whole embedded time series of selected neurons in the 10 selected samples is $X = [X_{,1}, X_{,2}, \ldots, X_{,10}] \in R^{(94 \times 300) \times (5701 \times 10)}$.

The cost function of Reconstruction ICA consists of the terms of the reconstruction cost and the independence (non-Gaussianity). Reconstruction ICA searches a matrix $W$ that minimize the cost function $\lambda ||WW^{\top}X^{\top} - X^{\top}||_2^2 + ||g(W^{\top}X^{\top})||_1 = \lambda ||WM - X^{\top}||_2^2 + ||g(M)||_1$, by using the L-BFGS optimization method implemented in Matlab, where $\lambda$ is the weight for the reconstruction penalty, $||\cdot||$ is the entrywise norm, and $g$ is the entrywise contrast function $g(x) = \frac{1}{2}log(cosh(2x))$. $M$ is the independent components $M = W^T X^T \in R^{n \times (94 \times 300)}$, and is regarded as the motifs of the neural activities. $W$ is the weight matrix $W \in R^{(5701 \times 10) \times n}$, and is regarded as the occurrences of the motifs. $n$ is the number of components. We repeated this Reconstruction ICA with different $n$ and visually inspected the results. Then we set $n$ as 14, which was the minimum number to capture the neural response to the sodium chloride stimulation.

For completing the missing values, the matrix factorization was combined with TDE-RICA. Matrix factorization does not take into account $M = W^T X^T$ and searches $M$ and $W$ that minimizes the reconstruction cost $||WM - X^{\top}||_2^2$. Therefore matrix factorization can be applied even if $X$ contains missing values. Here we extend the matrices included in the reconstruction cost. $X^{all}$ is the embedded time series of all 177 neurons in all 24 samples $X^{all} \in R^{(177 \times 300) \times (5701 \times 24)}$. $M^{all}$ is the extended matrix of motifs $M^{all} = [M, M^{new}] \in R^{n \times (177 \times 300)}$. $W^{all}$ is the extended matrix of occurrences $W^{all} = [W, W^{new}] \in R^{(5701 \times 24) \times n}$. Then we search $M^{new}$ and $W^{new}$ to minimize the extended reconstruction cost $||W^{all}M^{all} - X^{all,\top}||_2^2$ by using the L-BFGS optimization method implemented in Matlab.

## Synapse-based regression models

**gKDR.** To construct a model that reproduces the dynamics of the whole neural network, we adopted several approaches that model synaptic inputs to each neuron. First, we collect the presynaptic-postsynaptic relationships in the physical network. Connectivity data was adopted from the electron microscopy reconstruction data [7] which had been digitized by Oshio et al. (http://ims.dse.ibaraki.ac.jp/ccep/) [48]. Chemical synapses were treated as directional and gap junctions were treated as bidirectional synapses. In our 4D imaging data for each sample, there are many missing neurons, because 1) part of the nervous system can be obscure or too packed to obtain clear-cut signals for each neuron, 2) some neurons are difficult to name, for example because the positional relationship with surrounding neurons is atypical. Therefore, in one option, we included neurons directly connected to the target neuron (called "direct links") to

make the models, and in another option, we included neurons connected to the target neurons via two synapses in case the directly connected neuron was not observed (called "indirect links"). For the sake of simplicity, the neurons thus collected are denoted as "presynaptic neurons" for a given target neuron, even if the neurons are two synapses away in the case of employing indirect links.

To model dynamic relationships between the activity time-series of presynaptic neurons and that of target neurons, we generated regression models using time-delay embedding (S1 Text—File 23A). As for TDE-RICA, $x_{i,j}(t)$ depicts the activity of neuron $i$ at time $t$ in sample $j$. Each neuron was set as a target neuron one by one. For example, assume that neuron $i$ is set as a target neuron ($i = 1, 2, \ldots, M_j$; $M_j$ is the number of neurons in sample $j$ data set). Activity of target neuron $i$, $y_{i,j}(t+\Delta t) \equiv x_{i,j}(t+\Delta t)$, is predicted by activities of presynaptic neurons. Here, similar to TDE-RICA, time-delay embedding was adopted to utilize the past time series of the presynaptic neurons to predict target neuron activity. Namely, $y_{i,j}(t+\Delta t)$ is predicted from $\underline{X}_{i,j}(t) = [X_{\omega_{i,j,1},j}(t), X_{\omega_{i,j,2},j}(t), \ldots, X_{\omega_{i,j,M_{i,j}},j}(t)]$, where $X_{m,j}(t)$ stands for a row vector representing time-delay-embedded activity of neuron $m$, $[x_{m,j}(t - (k' - 1)\tau'), x_{m,j}(t - (k' - 2)\tau'), \ldots, x_{m,j}(t)]$. $\Omega_{i,j} = \{\omega_{i,j,1}, \omega_{i,j,2}, \ldots, \omega_{i,j,M_{i,j}}\}$ is a set of indices of neurons presynaptic to target $i$, $M_{i,j}$ being total number of neurons presynaptic to neuron $i$ (Fig 6A and S1 Text—File 23B). Note that previous activity of the target neuron itself, $X_{i,j}(t) = [x_{i,j}(t - (k' - 1)\tau'), x_{i,j}(t - (k' - 2)\tau'), \ldots, x_{i,j}(t)]$ was also included in the explanatory vectors $\underline{X}_{i,j}(t)$ ($\in R^{M_{i,j}+1}$).

Mean number of presynaptic neurons was 6.8 and 21.1, respectively, for direct links and indirect links (S1 Text—File 22). This causes a widely recognized challenge of the curse of dimensionality. Considering that synaptic transmission is an intrinsically nonlinear process, we employed the gradient kernel dimension reduction (gKDR) (32) for dimension reduction. The basic principle of KDR is explained in S1 Text—File 23. We shall model a function $y_{i,j} = f_{i,j}(\underline{X}_{i,j}) \simeq g_{i,j}(U_{ij})$ (here $y_{i,j}$, $\underline{X}_{i,j}$ and $U_{i,j}$ are considered random variables and $y_{i,j}(t)$, $\underline{X}_{i,j}(t)$ and $U_{i,j}(t)$ are samples from them), as a reasonably simple model, which is achieved by selecting a low dimensional subspace of explanatory variables as $U_{i,j}(t) = \underline{X}_{i,j}(t)B_{i,j}$, $B_{i,j}^T B_{i,j} = I_K$. $B_{i,j} \in R^{(M_{i,j}+1) \times K}$ is selected aiming at making $U_{i,j}(\in R^{1 \times K})$ sufficiently informative for predicting $y_{i,j}$. More precisely, gKDR evaluates statistical independence between $\underline{X}_{i,j}(t)$ and $y_{i,j}(t+\Delta t)$ conditioned on $U_{i,j}(t)$, using reproducing kernel Hilbert space (RKHS), and maximizes it. It thereby finds a subspace including $U_{i,j}(t)$ that is most informative for estimating $y_{i,j}$. In the toy example in S1 Text—File 23, the value of $y$ is shown in pseudo-colors. Although $y$ is a function of $x_1$, $x_2$ and $x_3$, $y$ depends only on the values of $u_1$ and $u_2$ and is independent of $u_3$ (an axis perpendicular to $u_1$ and $u_2$). Therefore, gKDR selects a $B$ that maps $(x_1, x_2, x_3)$ to $(u_1, u_2)$. The dimension of the subspace, $K$, needs to be pre-determined and in this work $K$ was optimized by a grid search as well as other hyperparameters $k'$, $\tau'$ and $\Delta t$ as described in the main text. gKDR was performed by Matlab codes distributed by Kenji Fukumizu on his web site (https://www.ism.ac.jp/~fukumizu/index_j.html).

**Sensory input.** As described above, we stimulated the animals with regular changes of salt concentrations while 4D imaging. In the gKDR-GMM model, salt sensor neurons were defined as ASEL/R, AWCL/R, BAGL/R and ASHL/R. ASEL and ASER are known to be salt-sensing neurons that are most important for salt chemotaxis. AWC and BAG were included because they are ciliated sensory neurons and consistently showed prominent activities synchronized to salt stimulus. ASH neurons were included because they have been shown to sense salt and occasionally showed salt response in our data set [49]. In the model, for these neurons, salt concentration was simply treated as an additional presynaptic neuron.

**gKDR-GMM.**   Determination of the dimension reduction matrix $B_{i,j}$ was achieved by gKDR as described in the previous section. In the case of gKDR-GMM, the mapping function $g_{i,j}(U_{i,j})$ is considered a probabilistic distribution. Here, rather than considering $g_{i,j}(U_{i,j})$ itself, the joint probability of $Z_{i,j} = (U_{i,j}(t), y_{i,j}(t+\Delta t))$ was modeled by using a Gaussian mixture model. Namely, $(Z_{ij}) \sim \sum_{q=1}^{\kappa}\{\pi_{i,j,q}N(Z_{ij}|\mu_{i,j,q}\Sigma_{i,j,q})\}$, $\sum_{q=1}^{\kappa}\pi_{i,j,q} = 1$. $\pi_{i,j,q}, \mu_{i,j,q}, \Sigma_{i,j,q}$ were determined by using $\{Z_{i,j}(t)\}$ from the training data. The optimal number of Gaussians ($\kappa$) was also searched for and as a result we employed two Gaussians as described in the main text. The fitgmm function of Matlab was used. Prediction by the gKDR-GMM was performed by random extraction from the conditional distribution $P(y_{i,j}(t+\Delta t)|U_{i,j}(t))$.

In our current implementation, processing 1000 time points required approximately 50–70 gigabytes of memory, and the computation time was roughly proportional to $O(n^2m)$, where $n$ is the number of time points, and $m$ is the number of variables, which is proportional to the number of presynaptic neurons. To provide an example, it took about 0.61 seconds per presynaptic neuron to develop a gKDR-GMM model for one target neuron or around 10–20 minutes for one animal using a high-performance 3110 TFLOPS supercomputer.

**gKDR-GP.**   Another popular nonlinear approach for probabilistic regression is Gaussian process. To see whether Gaussian process can be substituted for GMM, we also tested gKDR-GP, gKDR followed by Gaussian process. In this case, determination of the dimension reduction matrix $B_{i,j}$ was the same as above. Probabilistic distribution $y_{i,j}(t+\Delta t) \sim g_{i,j}(U_{i,j}(t))$ was modeled using Gaussian process. The fitrgp function of Matlab was used for this modeling. The results are shown in S1 Text—Files 10I-K and 24A-C.

**Cross validation and simulation based on the models.**   Prediction capacity of the gKDR-GMM model was evaluated by cross validation. Each time series data were evenly split into three parts, gKDR-GMM model was generated by one part, and by using the model, prediction was performed for each thirds of the data including the reserved two parts. Prediction was performed by using $\underline{X}_{i,j}(t)$ corresponding to the activities of presynaptic neurons up to time $t$, to predict the activity of the target neuron $y^*_{i,j}(t+\Delta t)$ ahead of time as a probability distribution; which was then compared to the real activity of the target neuron $y_{i,j}(t+\Delta t)$. Then mean log likelihood of the occurrence of target neuron's real activity $P(y_{i,j}(t+\Delta t)|U_{i,j}(t))$ was scored as above.

To see how gKDR-GMM model represents synaptic interactions, bootstrap test was performed as follows. The same cross validation was performed except that the presynaptic activity data were randomly permuted, keeping the activity of the target neuron itself unchanged. This was repeated 100 times, and the distribution of mean log likelihood as above was approximated by normal distribution and the one-sided p-value of the log likelihood of non-permuted data was estimated.

**Estimation of synaptic weights.**   To obtain a simplified estimate of synaptic weights, we took an average of the gradient, $\partial E(y_i)/\partial x_h$. for the gKDR-GMM model. $\partial E(y_i)/\partial u_g$ can be obtained by simple calculation of conditioned distribution of each Gaussian in GMM, and $\partial E(y_i)/\partial x_h = \sum_{g=1}^{K}(\partial E(y_i)/\partial u_g \partial u_g/\partial x_h)$. These values calculated for all $x(t)$ for the training data were averaged. $\partial E(y_i)/\partial x_h$ obtained for each lag $k$ were averaged with weight factor of $0.8^k$. This value was used as a "synaptic weight".

Consistency of synaptic weight estimation was performed by employing the idea of ensemble learning. gKDR-GMM models were generated with different Ks of $K = 3$, 4 and 5 using time series data with offsets of 0 to 4 as described below, and each time series were split into three. Therefore three models are made from three parts, for each offset. As a sum, 45 models were made for each target neuron in each sample. Of these, only those models that showed $p < 0.01$ for cross validation bootstrap test with both retained thirds of time series were used

as qualified. Synaptic weight estimation was calculated with each model as above for all qualified models. Then consistency of the weights were estimated by performing Wicoxon's rank sum test, to compare weight vector and a negative of the same vector. Based on these p values, FDR < 0.005 was used as criteria for consistent weights by the Benjamini-Hochberg method.

**Free-run simulation.** For simulation (also called free-run simulation), $y^*_{i,j}(t + \Delta t) = x^*_{i,j}(t + \Delta t)$ was predicted from $\underline{X}_{i,j}(t)$, and the prediction was repeated for all $i$'s, which allows for embedding to generate $X^*_{i,j}(t + \Delta t)$ and $\underline{X}^*_{i,j}(t + \Delta t)$ for all $i$, then $\underline{X}^*_{i,j}(t + 2\Delta t)$ was predicted from $\underline{X}^*_{i,j}(t + \Delta t)$ and this process was repeated using newly predicted values for further prediction.

**Virtual optogenetics.** To estimate the signal propagation through the neural network according to the gKDR-GMM model, virtual optogenetic stimulation was performed. For this test, only imaging data from samples 13 to 24 were used, where sensory stimuli by NaCl were not applied in the first half of the calcium imaging.

Activity of virtually stimulated neuron was set as a double-exponential function:

$$y = A\{-exp(-t/\tau_1) + exp(-t/\tau_2)\}$$

where $y$ is activity of the stimulated neuron, $t$ is time from initiation of stimulation, $A$ was determined so that $\max(y) = 2$, and $\tau_1 = 3$, $\tau_2 = 20$ were determined so that the time course profile mimics typical calcium responses of optogenetically activated *C. elegans* neurons. As a control experiment, activity of the same neuron was fixed to zero.

To mimic real optogenetic experiments, real activity profile of all observed neurons of the sample was used as initial values. Stimulation was initiated at several different time, separated by 400 time points, in the interval without NaCl input described above. After this time point, optogenetic stimulation or control stimulation was determined as above, and activity of all other neurons were determined as free-run simulation in the previous section. This simulation was repeated 10 times for each stimulation time.

To quantify the results, activity of each neuron during 80 simulation time steps (corresponding to approximately 80 s) after the initiation of optogenetic stimulation was averaged, and the resulting value for 10 repeats and different stimulation time points were all averaged. Finally, the values thus obtained from control simulation were subtracted from those from stimulated simulation.

For comparison with the results of [35], publicly shared codes and data accompanying the paper were downloaded from https://osf.io/e2syt/ and https://github.com/leiferlab/pumpprobe. Numerical data corresponding to the hue in Fig 2A and the number of valid observations in [35] was obtained by a modification of funatlas_plot_intensity_map.py and heat_raster_plot.py, respectively, provided at the GitHub site above.

**Hyperparameters.** Hyperparameter search was done by scoring similarity of correlation matrix of free-run simulation results to that of real data. Free-run simulation with a length of 2000 steps was repeated five times for each set of hyperparameters. Correlations of activity between neurons were represented by correlation matrices, and we scored similarity by calculating mean squared error of correlation matrices of real and simulation results. Although the data length is typically 6000 time points, data for every five time points were used for gKDR analyses because for gKDR, an inverse of $n \times n$ matrix needs to be calculated, $n$ being the data number, and using $n = 6000$ data points is impractical. For this reason, we have an option of selecting five phases of the data, starting from $t = 1$ (offset = 0), 2 (offset = 1) through 5 (offset = 4). Most of the analyses were done for offset 0, while the tests of consistency between different models or different samples were done using the data from all offsets.

Optimal hyperparameters for the gKDR-GMM model were searched for (S1 Text—File 10). First, presynaptic neurons were chosen as those that send direct chemical synaptic output to or have a gap junction with the target neuron (called direct links). However the reproduction of ensemble neural dynamics was not very good. This was expected because, as noted earlier, we cannot observe and annotate all neurons in the head of *C. elegans*, and typically, half of the neurons are missing from the data as described above (maximal number of annotated non-random neurons was 110 out of ~190 head neurons). Therefore, in another option, only for missing presynaptic neurons, neurons with another step of synaptic connections were included, namely presynaptic neurons for the missing neuron that was presynaptic to the target (called indirect links). The results of modeling including indirect links were generally better (S1 Text—File 10C-H; average numbers of "presynaptic" neurons in the connectome data, for direct link option and indirect link option were 15.1, 6.8 and 21.1, respectively; S1 Text—File 22). Next, embedding ranks and time steps was searched for and $k' = 30$ and $\tau' = 10$ was adopted (approximately 60 secs of time span was used for prediction of $y_{i,j}(t+\Delta t)$, S1 Text—File 10A and 10B). The total span of embedding is about the same as TDE-RICA, with wider spacing for gKDR. For $K$, namely the reduced number of dimensions after gKDR, the results were improved by increasing the dimension up to $K = 3$ or 4 but did not improve further (S1 Text—File 10C-E). Therefore, $K = 3$ to 5 was adopted. Because slightly different results were obtained with different $K$, we hereafter created models with these different values. The number of Gaussians were also varied and two Gaussians ($\kappa = 2$) showed the best performance in general, which we used hereafter (S1 Text—File 10F-H). The total span of embedding is about the same as TDE-RICA, with wider spacing for gKDR.

## Supporting information

**S1 Text. File information.**
(PDF)

## Acknowledgments

We thank Ms. Noriko Sato for Technical Assistance. The calculations were mainly performed in the Shirokane supercomputer at the Human Genome Center at the Institute of Medical Science, the University of Tokyo.

## Author Contributions

**Conceptualization:** Yu Toyoshima, Yuishi Iwasaki, Ryo Yoshida, Takeshi Ishihara, Yuichi Iino.

**Data curation:** Yu Toyoshima, Hirofumi Sato, Manami Kanamori, Moon Sun Jang, Takayuki Teramoto.

**Formal analysis:** Yu Toyoshima, Yuishi Iwasaki, Ryo Yoshida, Yuichi Iino.

**Funding acquisition:** Yu Toyoshima, Hirofumi Sato, Yuishi Iwasaki, Ryo Yoshida, Takeshi Ishihara, Yuichi Iino.

**Investigation:** Hirofumi Sato, Manami Kanamori, Moon Sun Jang, Koyo Kuze, Suzu Oe, Takayuki Teramoto.

**Methodology:** Yu Toyoshima, Takayuki Teramoto, Yuishi Iwasaki, Ryo Yoshida, Takeshi Ishihara.

**Project administration:** Yuichi Iino.

**Resources:** Manami Kanamori, Moon Sun Jang, Suzu Oe, Takeshi Ishihara.

**Software:** Yu Toyoshima, Daiki Nagata, Yuichi Iino.

**Supervision:** Yu Toyoshima, Takeshi Ishihara, Yuichi Iino.

**Validation:** Yu Toyoshima, Daiki Nagata, Yuichi Iino.

**Visualization:** Yu Toyoshima, Yuichi Iino.

**Writing – original draft:** Yu Toyoshima, Yuichi Iino.

**Writing – review & editing:** Yu Toyoshima, Hirofumi Sato, Daiki Nagata, Manami Kanamori, Moon Sun Jang, Koyo Kuze, Suzu Oe, Takayuki Teramoto, Yuishi Iwasaki, Ryo Yoshida, Takeshi Ishihara, Yuichi Iino.

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
