## [Decision Letter · Decision Letter 0]

25 Sep 2023

Dear Dr Iino,

Thank you very much for submitting your manuscript "Ensemble dynamics and information flow deduction from whole-brain imaging data" for consideration at PLOS Computational Biology.

As with all papers reviewed by the journal, your manuscript was reviewed by members of the editorial board and by several independent reviewers. In light of the reviews (below this email), we would like to invite the resubmission of a significantly-revised version that takes into account the reviewers' comments.

The reviewers raise important concerns related to the figures and methods that must be addressed. In addition, limitations of the methods in providing whole-brain imaging are critical to describe better.

We cannot make any decision about publication until we have seen the revised manuscript and your response to the reviewers' comments. Your revised manuscript is also likely to be sent to reviewers for further evaluation.

Sincerely,

Stacey D. Finley, Ph.D.

Section Editor

PLOS Computational Biology

Stacey Finley

Section Editor

PLOS Computational Biology

Reviewer's Responses to Questions

**Comments to the Authors:**

Reviewer #1: Unfortunately,I could not assess the paper as most of the figures (even when downloading the tiffs) had unreadable labels.

Reviewer #2: The Toyoshima et al. manuscript uses time-lapse 3D imaging of the whole nematode head and fits a phenomenological model that is constrained by the C. elegans connectome. The authors combined time-delay embedding and independendent compoenent analysis to derive the component dynamics (called "motifs") of the whole brain activities they recoreded from several animals. Then, the authors further constructed time series prediction models of synaptic communications by using dimension reduction and probabilistic modeling. The general idea of using computational modeling to investigate the functional and behavioral role of the connectome is good and timely. However, several points are technically unclear and, partly due to this, some results are weak. The authors need to discuss more openly the limitations of their imaging and their modeling methods and how they pertain to the interpretation of their results.

Major comments:

1. Most importantly, the authors never actually imaged the whole-brain for a period of time sufficient for their analysis. Line 351ff: “This was expected since approximately half of the neurons were missing from our data because of the difficulty in either observation or annotation (the maximum number of annotated non-random neurons was 110 out of ~190 head neurons).” This even makes the title which mentions “whole-brain” imaging inaccurate. “Whole-head” might be more accurate. Missing neurons cause major problems for a connectomically constrained “whole-brain” analysis and the authors work-around is to add presynaptic targets as indirect links to fill the void. The authors say “[…] this provided better results […]”. This is not surprising, since these indirect links add dynamically unconstrained extra parameters to the model, which will of course make any model fit better. Although I understand the limitations on the experimental methodogy, the authors need to explain clearly why they think they can make claims about “whole-brain” dynamics without imaging the whole brain.

2. The authors relegate the methods of the imaging methods almost entirely to reference 23. This is fine to focus the methods section on the analysis/model that is central to the paper. However, the authors should strongly consider including important aspects (those that pertain to the modeling in the manuscript) of the imaging into the present manuscript. For example, there is no mention of the temporal sampling frequency of the 3D imaging, which is critical for any discussion about dynamics. Even reference 23 is not entirely clear. It says there that “The exposure time was 500 ms”. I assume that is the exposure time for the entire cube and not just for a single plain, since the z-plane contains 50 voxels and 50*500ms would be too slow for any claims about dynamics.

3. On the critical role of noise: “Moreover, gKDR-GMM revealed the critical role of noise in realistic neural networks and enabled realistic simulation of whole-brain activities.” In biophysical models the biggest source of noise is from the activity of unknown presynaptic neurons. In a whole-brain model where the activity of every neuron is known, what do the authors believe is the source of the noise and are all noise sources going to be equally critical for network dynamics.

4. The NaCl gradient behavior seems to be happening on a particularly slow timescale and neurons that are not correlated with the behavior are likely excluded from the model because line 283f: “In these models, we only considered annotated neurons with non-random time series activity (time series with considerable autocorrelation).” The authors should discuss whether/how their results relate to the broader behavioral repertoire of C. elegans. To me, it seems that the results only apply to the specific behavior investigated. If that is the case this should be more clearly mentioned.

5. Line 59f: “Therefore, the network’s shape and synaptic properties are the primary determinants of information flow.” I somewhat disagree with this, since I would also consider intrinsic properties (and their changes at neuromodulator time-scales) primary determinants of information flow but I guess it depends what one considers “primary”. However, if the authors believe that their results pertain to the importance or irrelevance of intrinsic properties, I’d be interested to find a broader discussion of connectomics vs intrinsics in the manuscript.

6. In the time-delay embedding/ICA, I wonder whether and how the motifs of neuronal dynamics depend on the choice of a time window for these analyses. For example, some motifs (e.g., motifs 1 and 2) look like mirror images of half-cycles of an oscillatory activity pattern, but the detection of these motifs will depend on the choice of the window size. For instance, why does the method extract a complete oscillation cycle as a motif? Is the optimal window size for time-delay embedding autoatically determined by the method or chosen arbitrally by experimenters?

7. Line 268ff: "[...] This result suggests individual differences in the quantitative relationships of neurons between animals, despite the general assumption of little individual differences in neural activity in C elegans [...]" Are the individual differences also seen in animal behaviors? The statement does not make much sense to me without knowing the individual differences in neuronal activity and those in animal behavior. The former may be just a reflection of the latter.

8. Line 335ff: The gKDR-GMM model has two indices i and j, where i refers to neurons observed in samlpe j. This raises a doubt whether the gKDR-GMM model merely analyzes neuronal activty data from different samples (animals) in paralell. How does the method extract features commonly found across different samples? The current manuscript just describes the technical details of the analysis method, but the authors should also explain the conceptual ideas behind it in more detail. In addition, no quamtitative evaluation of the model's performance seems to be attempted in the manuacript, although I cannot deny the possiblity that I missed such information.

Minor comments:

9. Lines 637ff: “Therefore, there is a pressing need for new methods that offer truly quantitative understanding of the ensemble behaviors.” I find the wording here weird. In what sense are cross correlation and Granger casuality not “truly quantitative”? What does it mean to be truly quantitative? From the authors' mention of Granger Causality, I believe they are here expressing their desire for more causal inference methods in neuroscience as opposed to correlational (xcorr & Granger). If that is the case the authors should feel free to say so. If not, they should explain what makes a method “truly quantitative”.

10. It would be nicer if the distinction between group A and group B were more clearly visible in Fig. 8A-D. For instance, you may use colors for labeling the two groups.

Reviewer #3: Toyoshima et. al. present a powerful analysis of brain-wide calcium imaging data in mechanically restrained C. elegans. Authors have collected a unique dataset comprising annotated neural recordings across 24 animals. Their analysis pipeline is equally unique and commendable in my opinion for being non-parametric and comprehensive in scope. In the same study they have presented interpretable dynamic motifs using a thoughtful combination of delay embedding with reconstruction ICA and modeled the activity dynamics using a probabilistic kernel method to extract a functional connectome as well as neural transfer functions. It is also commendable that the authors have made their data available online. I also appreciated their systematic handling of the hyperparameters to separate the robust vs variable parts of their results.

After some modifications to the writing and overall presentation of their work (described below), I recommend this work for publication and expect it to become an exemplar of neural data analysis in large neural recordings.

I have following suggestions, most of which are related to the presentation of the work and putting it in context with existing literature.

1. I had a hard time reading the figures because of their low resolution. In Fig. 1 and 8, I couldn't read the neuron names. I also had a hard time seeing the top row in Fig 1., which has the stimulation protocol. These issues stand for all figures, but I felt was particularly important in Fig 1. and 8. Vectorizing the graphics and text would be preferable.

2. A schematic of their experimental setup along with the microfluidic chip would be good in Fig 1. I was under the impression that the worms were immobilized in the chip, but on line 241-242, authors say “This suggests that when worms move forward in the microfluidic device, their heads inevitably experience mechanical stimulation from the device’s wall.”

To the same point, some discussions about how much different animals were able to move in the microfluidic device would be appreciated. How much of the neural activity variance could be explained by the movement of the animals within the chip?

3. In my experience, the time-delay parameter has a big effect on the outcome. In this study, authors use a time-delay equivalent to the NaCl stimulation period (~60s). Which appears long to me, with this window, only the dynamics that are occurring at a minute long timescale are robustly identified, while faster dynamics are smoothed over. Usually, if there is a strong cycle in the data, a delay of about a quarter to a half of the cycle period is used (rule of thumb). I would have expected a delay parameter of the order of 30s to be used. There are systematic ways of optimizing the delay parameter through heuristics such as the autocorrelation timescale, predictability time (Ahamed et al. Nature Physics, 2021), predictive information (Costa et al. Chaos, 2023). It could be revealing to use these methods for optimizing the time delay parameter. If that's not possible, a demonstration of the robustness of motifs with different delay parameters could also be good.

4. In the result section on the motifs (lines 220-221): “Other neurons, including downstream interneurons, had lower weights, suggesting that activities of sensory neurons might only weakly influence the downstream neurons”. This is a potentially important point, and it would be nice if the authors can verify this. Not only will that add to the biology, but it will also give confidence that the motifs are interpretable. If the above is true, then the future activity of the downstream neurons cannot be predicted well by sensory neurons because it is only weakly influenced by them. Is it possible to test this point?

5. The statement on lines 397-398: “Upon surveying all results, it appears that there must be distinct groups of correlated neurons for the dynamics to be accurately reproduced”, needs be clarified a little. I would’ve expected that when the neurons exhibit globally correlated activity, their activity would be easier to reproduce. But it seems the opposite is true.

6. I like the analysis that the authors do to identify functional connections and time lags using their predictive model. It would be nice if the authors can compare their results (both the connections and lags/decay of responses) to the recent preprint by Randi et al. (bioRxiv 2023) from Andy Leifer’s lab, where an estimate of the functional connectome was presented by direct stimulation of neurons. How close does an estimate of based on predictive modeling of the dynamics get to an experimentally derived estimate?

7. The discussion is very thorough in its handling of the computational methods and various details, but I found the discussion lacking in its handling of the biology of the system. What are the key biological insights that were gained from the analysis? How reliable are they? What do they suggest about the system? Do they suggest any follow-up experiments? What are the main biological takeaways? These points are there in the paper (even in the abstract), but it would be nice to consolidate and expand upon them in the discussion.

**Have the authors made all data and (if applicable) computational code underlying the findings in their manuscript fully available?**

Reviewer #1: None

Reviewer #2: Yes

Reviewer #3: Yes

PLOS authors have the option to publish the peer review history of their article (what does this mean?). If published, this will include your full peer review and any attached files.

Reviewer #1: No

Reviewer #2: No

Reviewer #3: No
---

## [Decision Letter · Decision Letter 1]

21 Jan 2024

Dear Dr Iino,

We are pleased to inform you that your manuscript 'Ensemble dynamics and information flow deduction from whole-brain imaging data' has been provisionally accepted for publication in PLOS Computational Biology.

Best regards,

Matthieu Louis

Academic Editor

PLOS Computational Biology

Stacey Finley

Section Editor

PLOS Computational Biology

Reviewer's Responses to Questions

**Comments to the Authors:**

Reviewer #2: The authors have clarified my criticisms and revised the manuscript in a satisfactory fashion. So, I have no further comments and recommend the publication of the revised manuscript.

Reviewer #3: I thank the reviewers for their responses to my queries. I am satisfied with them and enjoyed reading the revised version of the manuscript. Fig S7 with a discussion of the time-delay parameter is appreciated and will give readers some guidance into how the choice was made. Fig 10 and Fig S21 are a welcome addition and make the study much stronger. It is non-trivial in my opinion to match the results of optogenetic experiments with the "virtual optogenetic" experiments performed through the model authors propose. This analysis provides confidence in both, the gKDR-GMM model and the experimental response maps produced by Randi et al. I enthusiastically support the article's publication.

**Have the authors made all data and (if applicable) computational code underlying the findings in their manuscript fully available?**

Reviewer #2: Yes

Reviewer #3: Yes

PLOS authors have the option to publish the peer review history of their article (what does this mean?). If published, this will include your full peer review and any attached files.

Reviewer #2: No

Reviewer #3: No

---

## [Editor Report · Acceptance letter]

19 Feb 2024

PCOMPBIOL-D-23-01239R1 

Ensemble dynamics and information flow deduction from whole-brain imaging data

Dear Dr Iino,

I am pleased to inform you that your manuscript has been formally accepted for publication in PLOS Computational Biology. Your manuscript is now with our production department and you will be notified of the publication date in due course.

With kind regards,

Judit Kozma
